# Synergistic effect of nano-selenium and metformin on type 2 diabetic rat model: Diabetic complications alleviation through insulin sensitivity, oxidative mediators and inflammatory markers

**Shaymaa A. Abdulmalek**[ID]**, Mahmoud Balbaa**[ID]*

Department of Biochemistry, Faculty of Science, Alexandria University, Alexandria, Egypt

* mahmoud.balbaa@alexu.edu.eg

**Data Availability Statement:** All relevant data are within the paper and its Supporting Information files.

## Abstract

### Background and objectives

In the present article, we explore a novel strategy of selenium nanoparticles (Se-NPs) for the treatment of type 2 diabetes mellitus (T2DM) by investigating the effect of Se-NPs alone and in combination with standard anti-diabetic drug metformin (MET) in high-fat diet/streptozotocin (HFD/STZ)-induced T2DM.

### Methods

HFD was supplemented daily to experimental rats for 8 weeks, followed by a single low dose injection of 35 mg/kg of STZ to induce T2DM. The synergistic effect of the different therapeutic strategies on diabetic complications was evaluated after the Se-NPs and MET administration for 8 weeks. Molecular and biochemical analyses were conducted to figure out the effectiveness of our treatment on insulin sensitivity, oxidative mediators and inflammatory markers.

### Results

Our observations demonstrated that HFD/STZ-induced rats have a toxic effect on serum and hepatic tissues resulted in inducing remarkable oxidative damage and hyper-inflammation with a significant disturbance in the insulin signaling pathway. Experimental animals either treated with mono-therapeutic-two doses Se-NPs (0.1 and 0.4 mg/kg) and/or MET (100 mg/kg) alone as well as the combined therapy resulted in a remarkable protective anti-diabetic effect illustrated by significant decreases in fasting blood glucose and insulin levels after 8 weeks treatment. At the same time, the levels of active insulin signaling proteins pIRS1/pAKT/pGSK-3β/pAMPK were significantly improved. Moreover, Se-NPs exhibited an anti-inflammatory effect by the mitigation of cytokine expression and a balance between oxidative stress and antioxidant status was restored. Furthermore, the anti-diabetic drug MET administration also exhibited a significant improvement in diabetic complications after the treatment period.

**Funding:** The authors received no specific funding for this work.

**Competing interests:** The authors have declared that no competing interests exist.

**Abbreviations:** AGE, advanced glycation end product; AHR, anti-hydroxyl radical; ALP, alkaline phosphatase; ALT, Alanine transaminase; AMPK, 5' adenosine monophosphate-activated protein kinase; AST, aspartate transaminase; BW, body weight; COX-2, cyclooxygenase-2; CTS, chitosan; CVP, catathelasma ventricosum polysaccharides; DM, diabetes mellitus; DMSO, dimethyl sulfoxide; DPPH, 1,1 diphenyl-2-picrylhydrazyl (H-A); GGT, Gamma-Glutamyl Transferase; GPx, glutathione peroxidase; GR, glutathione reductase; GSH, glutathione; GSK-3β, glycogen synthase kinase 3 beta; GST, glutathione-s-transferase; HDL, high density lipoprotein; HFD, high fat diet; iNOS, inducible Nitric oxide synthases; IL, interleukin; IR, insulin resistance; IRS, insulin receptor substrate; LDL, low density lipoprotein; LPO, lipid peroxidation; MDA, malondialdehyde; MET, metformin; MPE, mulberry leaf and *Pueraria Lobata* extracts; NF-κB, nuclear factor kappa-B; NO, nitric oxide; NSO, *nigella sativa* oil; PI3K, phosphoinositide 3-kinase; ROS, reactive oxygen species; RT, reverse transcriptase; RT-PCR, real-time polymerase chain reaction; Se-NPs, Selenium nanoparticles; STZ, streptozotocin; SOD, superoxide dismutase; T2DM, type 2-diabetes mellitus; TAC, total antioxidant capacity; TC, total cholesterol; TEM, Transmission electron microscopy; TG, triacylglycerol; TNF-α, tumor necrosis factor-alpha; XO, xanthine oxidase; Trx, treatment; ELISA, Enzyme-linked immunosorbent.

## Conclusion

This study provides mightily the mechanism of action of combined Se-NPs and MET as a promising therapeutic alternative that synergistically alleviates most of diabetic complications and insulin resistance.

## Introduction

Over the past few years, the worldwide epidemic chronic diseases like T2DM have attracted attention. The prevalence of diabetes was 425 million in 2017 according to the International Diabetes Federation (IDF), this number is anticipated to rise by 2045 to 629 million with a 48% increase. In addition, the incidence of diabetes in the Middle East and North Africa was 39 million in 2017 and it will rise to 82 million by 2045 with a 110% increase. Among all types of diabetes, T2DM accounts for the vast majority of diabetes cases accounting for around 90% (IDF Diabetes Atlas. 2017). Many studies have the aim to clarify the metabolic and molecular alterations in the insulin signaling pathways arise in T2DM and involved in the development of IR, one of the major hallmarks of T2DM pathogenesis [1]. Clearly, the insulin signaling pathway starts by binding of insulin to its receptors on target cells. Then the activation events begin in the cells by activation of insulin receptor-β and IRS1, thereby recruiting PI3K to its location. The main target of PI3K in hepatic cells is AKT, which plays a key role in glucose uptake [2]. Previous studies demonstrated that the up-regulation of pIRS1 and pAKT play a major role in improving glucose uptake and regulating its blood levels [3].

Supportive studies have established that oxidative stress is implicated in the pathogenesis of T2DM and represent a risk factor in IR [4]. Besides, hyperglycemia leads to free radical production by glucose oxidation, non-enzymatic glycation of proteins and elevation of lipid peroxidation [5]. This exaggerated ROS production can reduce the antioxidant enzyme activities directly. Alongside with oxidative stress and hyperglycemia, hyperlipidemia has been prescribed as a causative factor for augmenting morbidity and mortality in diabetic individuals [6]. Moreover, various inflammatory mediators arise from activation of stress-signaling pathways that interlinked with IR by diverse molecular mechanisms. The chronic release of pro-inflammatory markers such as TNF-α, IL-6 induce IR by blocking insulin signaling protein activation [7, 8]. In addition, diverse factors participate in the propagation of IR, so repression of these causative factors is the key to prevent and treat T2DM. We speculate that finding efficient multifunctional therapeutic agents act more broadly to repress diabetic complications may offer a greater advantage. Accordingly, several studies have been focusing specifically on minerals due to their interference in the synthesis of insulin as well as regulation [9]. Nowadays, several nanoparticles used as alternative therapy due to their multifunctional biological activities, which treat diabetic complications and combat inflammation. Interestingly, selenium is an essential antioxidant and anti-inflammatory micronutrient that can prevent ROS production by improving the activity of GPx and selenoproteins [10]. Additionally, Se-NPs have their unique biological advantages with excellent high bioavailability and 7-fold lower toxicity than sodium selenite [11]. The biomedical application of Se-NPs demonstrated its antioxidant properties [12]. Besides its unique abilities, for more stability of Se-NPs in solutions, stabilizing agents such as polysaccharides are added; the active hydroxyl groups in polysaccharides can improve the bioavailability and biological activity producing a synergistic effect between Se-NPs and polysaccharides [13]. Interestingly, Se contributes to a reduction of fasting serum insulin and IR index, which resulted from its behavior as insulin-like action.

Moreover, a recent study reported that Se-NPs were found to inhibit MAPK, NF-κB and TNF-α levels in rats [14, 15].

Diabetic patients shared several pathological features comprising inflammation and oxidative stress. Owing to the efficient anti-inflammatory and antioxidant actions of selenium, several studies revealed the link between DM and serum level of selenium. Previous studies have shown that the higher selenium level is linked with a lower risk of T2DM development [16, 17, 18]. Interestingly, Se-NPs have an intrinsic hypoglycemic effect beside its antioxidant and anti-inflammatory activities, so T1DM and T2DM can be treated with Se-NPs through alleviation of oxidative stress and sensitizing insulin [19].

Preceding studies demonstrated the hypoglycemic effect of Se-NPs in STZ-induced T1DM that showed a significant diminished in glucose levels with augmentation of insulin level and normalization of the liver and renal function combined with enhancement of lipid profile [20]. Furthermore, Liu et al., illustrated the administration of CVPs-Se-NPs ameliorated the blood sugar, body weight, oxidative stress and dyslipidemia in STZ-induced diabetic mice [21]. Additionally, CTS-Se-NPs at dose 2 mg/kg exhibited a strong anti-diabetic activity in STZ-induced diabetic mice by enhancing body weight, decreasing glucose and insulin levels, decreasing serum lipid profile, and improving antioxidant status [22]. Moreover, the treatment of STZ-induced rats by liposomal-Se-NPs for 21 days results in a marked decline in glucose level with a marked increase in serum insulin and pancreatic antioxidant [23]. The use of Se-NPs for oral delivery of insulin at a dose of 25 IU/kg for 2 weeks showed a superior hypoglycemic effect that it could alleviate oxidative stress, restore the impaired β-cell function of the pancreatic islet and promote glucose utilization. In addition, the bioavailability of Ins-Se-NPs was up to 9.15%, which was significantly higher than that of the insulin solution [24]. A recent study revealed that the oral administration of MPE-Se-NPs produced a marked hypoglycemic effect both in normal and diabetic rats. It could attenuate oxidative stress by a significant decline of all peroxides and a significant increase in all antioxidants. It improves pancreatic function by the dramatic restoration of impaired β-cell. It promotes glucose utilization by adipocytes. In addition, oral MPE-Se-NPs bioavailability was enhanced depending on the excellent intestinal permeability and transepithalial transport [25]. From all the above studies, Se-NPs based nanomedicines have shown the great importance of the application to diabetes care.

In the present study, we prepared Se-NPs with a one-step method by reducing sodium selenite with ascorbic acid. To prevent the aggregation of the particles and to improve the stability, dextrin has been introduced into the redox system. Owing to their high bioavailability, low toxicity, and novel therapeutic properties, Se-NPs have been recognized as a promising tool for drug therapies in T2DM. In this study, we designed new therapeutic strategies for T2DM to clarify the potential role of the administration of two doses of Se-NPs and/or Se-NPs combined with MET in order to investigate whether Se-NPs can show anti-diabetic activity, or even improve the therapeutic effect *in vivo*. Also, we evaluate the modulatory role of Se-NPs in amelioration of oxidative damage in the hepatic tissue of HFD/STZ-diabetic rats as well as restoring antioxidant defense capacity. Further, we examine whether oral administration of Se-NPs and MET can delay the onset of IR, possibly by affecting the insulin signaling pathway and the inflammatory pathway.

## Material and methods

### Materials and chemicals

TRIzol RNA Isolation Reagents (Catalog No 15596026, Invitrogen). EXPRESS One-Step SYBR GreenER Kit, universal (Catalog No 11780200, Invitrogen). DEPC-treated water (Catalog No

D5758), Triton X100 (CAS No 9002-93-1), STZ (white to yellow powder, CAS No 18883-66-4). HEPES, Trypan Blue, and MTT (Sigma-Aldrich, USA). Fetal Bovine serum, RPMI-1640, HEPES buffer solution, L-glutamine, and gentamycin (Lonza, Belgium). The primer sequences (dried in 2 ml screw cap tube) were obtained from Sigma-Aldrich, USA. Antibodies included, [anti-AKT1 (NBP2-01724) or anti-AKT1-p$^{ser473}$ (NBP2-35349)], [anti-GSK-3β (MAB2506) or anti-GSK-3β-p$^{Ser9}$ (NB100-81948)], [anti-IRS1 (NB100-82001) or anti-IRS1-p$^{Tyr612}$ (NBP1-73967)], [anti-AMPK alpha (MBS835324)], [anti-AMPK-alpha-p$^{T172}$ (MBS462009)] [anti-p65-p$^{Ser536}$ (NB100-82088)], [anti-COX2 (NB100-689)] and [anti-β-actin (NB600-501)]. TNF-α (Catalog No: MBS355371), iNOS (Catalog No: MBS723326). IL-6 (Catalog No: MBS355410), IL-1β (Catalog No: MBS825017), AGEs (Catalog No: MBS774145) and Rat Insulin (Catalog No: MBS760915) assay kits were supplied by Mybiosource (San Diego, CA, USA). MET tablet (500 mg) was purchased from Eva Pharma, Egypt. Solvents and other associated biochemical reagents including sodium selenite, sodium dodecyl sulfate, Tween 20, and other commonly used reagents were obtained from with high grade from Sigma-Aldrich, USA.

## Preparation of Se-NPs

Synthesis of Se-NPs was performed according to the method described by Qian Li et al. [26] with some modifications. Sodium selenite (100 mM) and ascorbic acid (50 mM) stock solutions were prepared. The reacted sodium selenite to ascorbic acid ratios were varied (1:1, 1:2, 1:3, 1:4, 1:5 and 1:6) from the stock solution. Under magnetic stirring, the ascorbic acid solution was added dropwise to the solution of sodium selenite for 30 min at room temperature. Then the mixtures were allowed to react till the observation of color changing from colorless to light red. After that, the mixture was diluted to 25 ml using Milli-Q water.

## Coating of Se-NP

The Se-NP prepared previously were coated with Dextrin at a concentration (5%), which added after the appearance of color using a magnetic stirrer at room temperature using a single layer coating method. Prepared nanoparticles were diluted using the dextrin solution instead of adding water. Finally, nanoparticles were washed and dried using lyophilizer. The prepared nanoparticles were characterized using TEM.

## Transmission Electron Microscope (TEM)

Se-NPs were prepared for TEM analysis by placing a drop of the nanoparticle suspension on carbon-coated copper grids. Under an infrared lamp, the samples were dried and then the images were recorded using TEM Philips CM 200 instrument with an operating voltage at 80 KV and resolution of up to 2.4 A˚.

## Cytotoxicity study

**Cell line and maintenance.** Sprague-Dawley male rats (200–250 g) were obtained from the animal house of the Medical Technology Center, Alexandria University. Hepatocyte isolation was performed according to the collagenase perfusion described by Reese and Byard [27]. Hepatocytes ($1 \times 10^6$ cells/ml) were placed into Krebs-Henseleit buffer (pH, 7.4) containing 12.5 mM HEPES and kept at 37˚C with 95% $O_2$ and 5% $CO_2$. Hepatocytes with a viability of more than 90%, which was measured with Trypan Blue were used in the experiments [28].

**Cytotoxicity assay.** For cytotoxicity assay, the hepatocytes were grown on RPMI-1640 medium and supplemented with 10% inactivated fetal calf serum and 50 μg/ml gentamycin at a concentration of $2x10^6$ cell/well in Corning 96-well tissue culture plates. The cells were

incubated for 24 hours at 37˚C in a humidified atmosphere with 5% $CO_2$ and suspended in the medium at a concentration of $2x10^6$ cell/well in Corning 96-well tissue culture plates. The prepared Se-NPs were then added into the 96-well plates (three replicates) to achieve eight concentrations (1000, 500, 250, 125, 62.5, 31.25, 15.6, 7.8 μg/ml). Six vehicle controls were run for each 96 well plate as a control. After incubating for 48 h, the numbers of viable cells were determined by the MTT test. Briefly, the media was removed from 96 well plate and replaced with 100 μl of fresh culture RPMI 1640 medium without phenol red, and then 10 μl of the 12 mM MTT stock solution (5 mg of MTT in 1 ml of PBS) were added to each well including the untreated controls. The 96 well plates were then incubated at 37˚C and 5% $CO_2$ for 4 hours. An 85 μl aliquot of the media was removed from the wells, and 50 μl of DMSO was added to each well and mixed thoroughly with the pipette and incubated at 37˚C for 10 min. Then, the optical density was measured at 590 nm with the microplate reader (SunRise, TECAN, Inc, USA) to determine the number of viable cells and the percentage of viability. The percentage of cell viability was calculated by the formula: % Cell Viability = [(At/Ac)] x 100%, where At is the mean absorbance of wells treated with the tested sample and Ac is the mean absorbance of untreated cells.

### *In vitro* antioxidant bioactivities of Se-NPs

**DPPH radical scavenging activity.** DPPH assay was proceeded to show the free radical scavenging activity of the synthesized Se-NPs using the method described by Patel Rajesh and Patel Natvar [29] with some modifications. The DPPH solution was added to the serial concentrations of synthesized Se-NPs and standard ascorbic acid after preparations (0.25, 0.5, 1, 1.5 and 2 mg/ml) and diluted with methanol. After 15 min the absorbance was read at 517 nm using methanol as the blank. Also, for a control reading, 150 μl of DPPH solution was added into 3 ml of methanol and the absorbance was taken immediately at 517 nm. The calculation of DPPH radical scavenging activity was carried out using the following formula:

% scavenging $= \frac{A0-A}{A0} * 100$, where $A_0$ is the absorbance of the control and A is the absorbance of the test sample.

**NO radical scavenging activity.** NO scavenging activity was carried out according to the method of Sreejayan and Rao [30]. Serial concentrations of synthesized Se-NPs, as well as ascorbic acid (standard) 0.25, 0.5, 1, 1.5 and 2 mg/ml, were prepared and dissolved in DMSO and 2.0 ml of sodium nitroprusside (10 mM) in phosphate buffer saline was added to each and incubated for 150 min at room temperature. After the incubation, 5 ml of Griess reagent was added to all tubes, including the control. The absorbance was measured at 546 nm on the UV-Visible spectrophotometer; methanol was used as the blank. The percentage of the scavenging activity was calculated as explained above.

**Hydrogen peroxide scavenging activity.** The ability of Se-NPs to scavenge hydrogen peroxide was examined spectrophotometrically [31]. Different concentrations of synthesized Se-NPs, as well as ascorbic acid (standard) 0.25, 0.5, 1, 1.5 and 2 mg/ml, were added to 0.6 ml of hydrogen peroxide solution (2 mM) prepared in phosphate buffer pH,7.4. Absorbance was measured against a blank solution (phosphate buffer) at 230 nm and was compared with the Ascorbic acid (standard). Hydrogen peroxide scavenging effect percentage was calculated as explained above.

**Reducing power assay.** The reducing power of Se-NPs was determined by the method of Yildirim et al. [32]. Various concentrations of synthesized Se-NPs, as well as ascorbic acid (0.25, 0.5, 1, 1.5 and 2 mg/ml) were mixed with 2.5 ml phosphate buffer (0.2 M) and 2.5 ml potassium ferricyanide (1%). The mixture of Se-NPs and ascorbic acid were incubated for 20 min at 50˚C. After cooling, 2.5 ml trichloroacetic acid (10%) was added to the mixtures and

centrifuged for 10 min at 3000 rpm. 2.5 ml of the supernatant was mixed with 0.5 ml freshly prepared $FeCl_3$ (0.1%). Then the absorbance was measured at 700 nm. The reducing power assay percentage was calculated as explained in the DPPH assay.

**Total antioxidant capacity assay.** The TAC assay of Se-NPs was determined by a standard method of Umamaheswari and Chatterjee [33]. Different concentrations of Se-NPs and ascorbic acid (0.25, 0.5, 1, 1.5 and 2 mg/ml) were added to 1.0 ml of the solution containing sulphuric acid (0.6 M), sodium phosphate (28 mmol) and ammonium molybdate (4.0 mmol). The mixtures were incubated for 90 min at 95°C. The absorbance was measured after cooling at 695 nm. The inhibition was calculated as explained in the DPPH assay.

## Experimental animals

The study was conducted on adult male Wistar rats (*Rattus norvegicus*, 90–120 g) in twelve different groups (each containing ten animals). The rats were obtained from the animal house of the Medical Technology Center, Alexandria University and fed *ad libitum*. All animal studies were performed according to the animal protocols approved by the Institutional Ethics Committee of Faculty of Science, Alexandria University, Egypt in accordance with the ethical standards. Rats were chosen after a behavioral and physical examination. The animals were housed in standard conditions of our laboratory in polypropylene cages, supplied with water and food in the animal houses. The animals were retained in a clean environment with 12:12 h light/dark cycle. The air was conditioned at 22±3°C with maintaining the relative humidity between 30–70% with a 100% exhaust facility. T2DM in rats was induced by supplementation of freshly prepared HFD for 8 weeks (17% carbohydrate, 58% fat, 25% protein, 253 g/kg casein, 310 g/kg butter, 10 g/kg cholesterol, 1.0 g/kg yeast powder, 60 g/kg vitamins and minerals, and 1.0 g/kg sodium chloride). The pre-diabetic stage, hypertriglyceridemia, and hypercholesterolemia were confirmed by elevated serum TC and TG as well as glucose. After 8 weeks a single intraperitoneal injection of low dose freshly prepared STZ (35 mg/kg) diluted in sodium citrate buffer (100 mM), pH 4.5 was given to the overnight fasted rats [34]. To recognize blood glucose levels in HFD/STZ-induced rats, every three days a drop of blood was taken from rats and examined by using Accu-check blood glucometer (Roche Diagnostics, Basel, Switzerland). HFD/STZ-diabetic rats showed blood glucose concentration higher than 300 mg/dl three days after the STZ injection were considered hyperglycemic.

## Experimental design

Six control non-diabetic groups receiving a normal diet were divided as follows:

Control (received physiological saline), Se-NPs (daily administered Se-NPs 0.1 mg/kg BW), Se-NPs (daily administered Se-NPs 0.4 mg/kg BW), MET (daily administered MET 100 mg/kg BW), Se-NPs-MET (daily administered Se-NPs 0.1 mg/kg and 100 mg/kg BW) and Se-NPs-MET (daily administered Se-NPs 0.4 mg/kg and 100 mg/kg BW). All doses were administered orally to experimental rats using a needle fitted to a disposable syringe [35]. After the verification of type 2 diabetes induction, the diabetic rats were classified as described in Fig 1 to HFD/STZ (untreated group), Se-NPs (0.1 mg/kg BW) group, Se-NPs (0.4 mg/kg BW) group, MET (100 mg/kgBW) group, Se-NPs-MET (0.1 mg/kg or MET 100 mg/kg), and Se-NPs-MET (0.4 mg/kg or MET 100 mg/kg) group. All animals were checked for any behavioral changes and abnormal clinical signs daily. After 8 weeks of daily oral administration, overnight fasting animals were weighed, anesthetized with sodium pentobarbital (i.p., 100 mg/kg) to minimize animal distress and suffering, blood was collected using a syringe puncture from the abdominal aorta and plasma and serum were isolated. The liver tissues were rapidly excised, washed with ice-cold 0.9% NaCl, frozen in liquid nitrogen and stored at -80°C for subsequent analyses.

**Fig 1. Experimental design for the study of HFD/STZ-induced T2DM in rats post-treatment with Se-NPs and MET monotherapy and combined therapy.**

## Body weight analysis

All the rats were weighed every 7 days after HFD supplementation in the induced groups as well as normal diet-control groups. The average BW gain was determined and recorded.

## Biochemical analyses

The assays of blood glucose and serum insulin levels were performed. The functioning of the liver was studied by analyzing liver enzymes such as ALT, AST, ALP, and GGT, as well as liver functional proteins, Albumin, total Bilirubin, and total protein. The functioning of the kidney was studied by measuring levels of creatinine, urea, and uric acid. The serum lipid profile was investigated by HDL-c, LDL-c, TC, total lipid and TG. The biochemical serum parameters were analyzed using an automatic biochemical analyzer based on the instructions supplied with the commercial assay kits.

## Assessment of insulin resistance and β-cell function

Homoeostasis model assessment (HOMA) β-cell function (HOMA-β%) and of IR (HOMA-IR) were carried out from fasting blood glucose (mg/dl) and fasting serum insulin (mM) by the HOMA method [36] using the following equations: IR (HOMA-IR) = [fasting glucose (mg/dl) × fasting insulin (μIU/ml)]/405, and β-cell function (HOMA-β) % = [360 × fasting insulin (μIU/ml)]/(fasting glucose (mg/dl)– 63). The insulin was performed by ELISA kit following the manufacturer's instructions.

## Oxidant and antioxidant assays

Serum and tissue oxidants and antioxidant enzymes activities were performed such as total-SOD, CAT, GPx, GR, GST, and XO were tested along with non-enzymatic antioxidants such as GSH, TAC, and AHR. Beside, lipid peroxidation in the liver and serum was determined by MDA assay. Meanwhile, the level of NO in the liver homogenate and serum was determined. In brief, the LPO assays were assessed according to the previous method and MDA concentrations were expressed as nmol/mg protein [37]. The NO assay was assessed as described before and expressed as μM/mg protein [38]. The GSH assay, non-enzymatic antioxidant was performed as the previous assay and expressed as mg/mg protein [39]. GST, GPx and total-SOD assays were carried out according to the previous assay and their activities were expressed as μmol/min/mg protein, nM/min/mg protein and μg/min/mg protein, respectively [40, 41, 42]. XO assay was carried out and its activity was expressed as μmol/h/mg protein [43]. The CAT and GR activities were assessed and their activities were expressed as μg/min/mg protein

and µmol/min/mg protein, respectively [44, 45]. AHR was assessed by using a commercial kit following the manufacturer's protocol. The liver tissues of each group were excised and homogenized in ice-cold PBS, pH 7.4, (10 ml/gm) liver tissue and centrifuged at 6000 rpm for 20 min at 4°C. The supernatant was collected and subjected for the determination of the oxidant and antioxidant biomarkers in liver tissues. The total protein content of the liver homogenate was analyzed by the standard protocol described by Lowry et al. [46].

## Determination of inflammatory cytokines by ELISA

Frozen liver tissues were homogenized in lysis buffer (150 mM NaCl, 1% Triton X-100 and 10 mM Tris, pH 7.4) containing a protease inhibitor, the liver homogenate was centrifuged at 6000 rpm for 20 min at 4°C and the supernatant was then collected for examinations of ELISA markers. The protein level of the liver lysate was measured using the method of Lowry et al. [46]. Serum samples and liver lysate were analyzed for levels of inflammatory cytokines, including TNF-α, IL-6, IL-1β, iNOS and AGEs using commercially available ELISA kits, according to the manufacturer's instructions.

## RNA isolation and quantitative real-time-PCR analyses

Total RNA was isolated from frozen liver tissues using a TRIzol RNA Isolation Reagents. Total RNA was quantified using a NanoDrop 2000/2000c spectrophotometer. The isolated RNA was used as a template together with each primer to carry out real-time PCR reactions using the EXPRESS One-Step SYBR GreenER Kit. The typical thermal profile for the PCR reaction 50°C for 5 min (cDNA synthesis) and then 95°C for 2 minutes, followed by 40 cycles of 95°C for 15 s, 60°C for 1 min and 72°C for 1 min. The relative quantity of each gene in the liver tissues of rats was normalized to β-actin and the relative expression fold change was calculated as described before [47]. The log value of $2^{\Delta\Delta Ct}$ is 2, where ΔΔCt was calculated by subtracting the β-actin cycle threshold value (Ct) from each of the target gene Ct. Then the calculation of ΔΔCt carried out by subtracting the resulted ΔΔCt values from ΔΔCt mean value of the control. The PCR primers for the following genes were considered:

IL-23 sense: 5′-CAT GCA CCA GCG GGA CAT ATG- 3′, anti-sense 5′-CAG ACC TTG GCG GAT CCT TTG-3′. IL-1α sense: 5′-CCC GAC TGC CTG CTG CTT CTC-3′, antisense: 5′-GAT CTG CCG GTTTCT CTT AGT CA-3′. MCP-1 (monocyte chemoattractant protein-1) sense: 5′- CATGCTTCTGGGCCTGCTGTTC-3′, antisense 5′-CCT GCT GCT GGT GAT CCT CTT GTA G-3′. socs-3 (suppressor of cytokines signaling) sense: 5′-ACT TGT TTG CGC TTT GAT TTG GTT T-3′, antisense: 5′-GTT GGG CAG TGG GAG TGG TTATTT-3′. Ox-LDL (oxidized low-density lipoprotein receptor-1) sense: 5′-GCC TCC TGT TGC TGC ATG AAA G- 3′, antisense 5′-CTCGGACGAGCT TTGCCT TTG-3′. β-actin sense: F:5′-GCCATGTACGTAGCCATCCA-3′, antisense: 5′-GAACCGCTCATTGCCGA TAG-3′ [48].

## Western blotting analysis

Frozen liver tissues were homogenized in ice-cold lysis buffer (20 mM Tris–HCl (pH, 7.5), 150 mM NaCl, 1% Triton-X 100, 1 mM EDTA) containing protease inhibitor cocktail. The homogenates were centrifuged at low speed at 2000 g for 10 min at 4°C and the supernatants were collected for measurement of protein concentrations and Western blot. Briefly, after separation on a 12% polyacrylamide gel, proteins were transferred to nitrocellulose membranes and then incubated using specific primary antibodies to figure out the target proteins. The following antibodies were used: anti-AKT1 or anti-AKT1 p$^{ser473}$, anti-GSK-3β or anti-GSK-3β p$^{Ser9}$, anti IRS1, anti-IRS1 p$^{Tyr612}$, anti-p65-p$^{Ser536}$, anti-COX2, anti-AMPK-α, anti-AMPK- α

$p^{T172}$ and anti-β-actin immune-blots were carried out on the prepared lysate according to the preceding method [49]. After the overnight reaction of the primary antibody at 4˚C, the membrane was washed three times with TBST (2.7 mM KCl, 137 mM NaCl, 19 mM Tris base, and 0.1% Tween 20) and incubated with secondary antibodies for 1 h at room temperature. Finally, the membrane was washed with TBST, The bands were detected by enhanced chemiluminescence (ECL detection kit) and band densities were quantified by using Image J program.

### Statistical analysis

Values were expressed by means±SE (standard error of the mean). Data were statistically analyzed by one-way analysis of variance (ANOVA). Duncan's test was used as a post hoc test for the comparison of significance between groups. P values less than 0.01 were considered significant.

## Results

### Preparation and characterization of Se-NPs

Se-NPs were designed in the redox reaction system of sodium selenite / ascorbic acid and finally coated with dextrin Se-NPs. The particle size and shape of the Se-NPs were determined by TEM, as shown in (Fig 2). Most of the Se-NPs had a diameter between 30 and 80 nm and the nanoparticles had a spherical shape.

### Cytotoxicity analysis

The biosynthesized Se-NPs were tested against the rat hepatocytes cell line. The tested Se-NPs exhibited potent activity in the tested hepatocytes. As shown in Table 1, the Se-NPs were not able to have a marked effect on the viability of hepatocyte cells on increasing their concentration and hence Se-NPs are non-toxic for this type of healthy cells.

### Antioxidant activity of Se-NPs

During cellular metabolism, various free radicals are generated, which cause damage to living organisms. The balance between generated free radicals and antioxidant status in the cells may terminate the oxidative potency by scavenging the free radicals. In the current study, the DPPH free radical scavenging efficiency of synthesized Se-NPs increased with increasing concentrations of a nanoparticle. The Se-NPs manifested maximum scavenging power of 81%

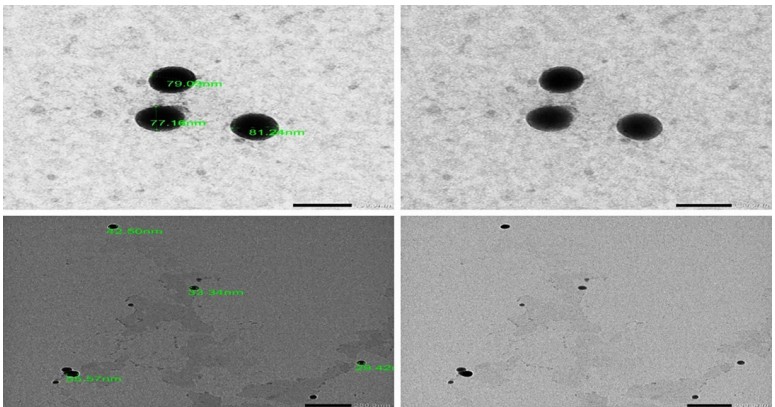

**Fig 2. Representative transmission electron microscopy analysis of prepared Se-NPs showing their size and shape.**

**Table 1. Evaluation of cytotoxicity of Se-NPs against Rat hepatocytes cell line by MTT assay.** Values are expressed as mean ± SE.

| Sample concentration (µg/ml) | Mean Viability% | Inhibitory % |
|---|---|---|
| 0 | 100 | 0 |
| 7.8 | 100 | 0 |
| 15.6 | 100 | 0 |
| 31.25 | 100 | 0 |
| 62.5 | 100 | 0 |
| 125 | 99.85±0.19 | 0.15 |
| 250 | 98.76±0.42 | 1.24 |
| 500 | 93.41±0.63 | 6.59 |
| 1000 | 91.4±1.36 | 8.6 |

(Fig 3A), similarly, NO scavenging activity displayed a maximum activity of Se-NPs of 84% (Fig 3B). The hydrogen peroxide scavenging activity also increased with increasing concentrations of the Se-NPs and showed maximum power of 77% (Fig 3C). TAC and reducing power assay showed a maximum level of 89% (Fig 3D) and 83% (Fig 3E), respectively. Because of this scavenging potency of Se-NPs, they are used in the management of chronic diseases such as diabetes.

## Changes in body weight

HFD intake was similar for all the induced and treated groups throughout the intervention period, but variations in weight gain were observed between the HFD/STZ-induced group and other treated groups at the end of 16 weeks (Table 2). After HFD intake, the highest weight gain was observed in all induced groups between the beginning of the experiment and at week 8 that showed obesity characteristics. After STZ administration, rats showed a decrease in the body weights from the first week of STZ injection. On the other hand, a BW gain was observed in HFD/STZ groups after administration of two doses Se-NPs and/or MET monotherapy and combined therapy for 8 weeks (Fig 4A). Moreover, control groups that received a normal diet showed a marked regular increase in their BWs during the 16 weeks (Fig 4B).

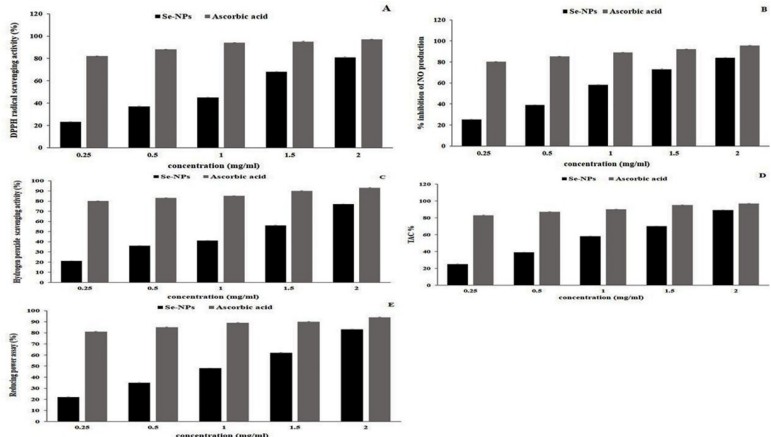

**Fig 3. Dose-dependent antioxidant activities of Se-NPs.** The antioxidant activity was measured by five different assays for each parameter. A: DPPH radical scavenging activity, B: NO free radical scavenging activity, C: Hydrogen peroxide scavenging activity, D: total antioxidant activity, E: Reducing Power assay.

**Table 2. Effect of two doses of Se-NPs, MET monotherapy, and combined therapy on the blood glucose levels, serum insulin levels, HOMA-IR, HOMA-β, and final body weight over 16 weeks in control and HFD/STZ-experimental rats.**

| Groups | Glucose (mg/dl) | Insulin (μIU/ml) | HOMA-IR | HOMA-β | Final BW (g) |
|---|---|---|---|---|---|
| Control | 95±2.2 [b] | 11.88±0.22 [b] | 2.78±0.12 [b] | 133.65±1.12[b] | 150±2.4 [b] |
| Se-NPs (0.1mg) | 92±2.4 [b] | 12.7±0.24 [b] | 2.88±0.14 [b] | 157.65±1.4 [a,b] | 145±2.6 [b] |
| Se-NPs (0.4 mg) | 90±3.4 [b] | 12.7±0.25 [b] | 2.82±0.12 [b] | 169.33±1.6 [a,b] | 155±2.89 [b] |
| MET | 87±2.3 [b] | 11±0.199 [b] | 2.36±0.11 [b] | 165±1.55 [a,b] | 135±3.5 [b] |
| Se-NPs (0.1mg)-MET | 82±2.7 [a,b] | 11.3±0.14 [b] | 2.28±0.13 [b] | 214.10±2.1[a,b] | 125±2.5 [b] |
| Se-NPs (0.4 mg)-MET | 80±3.5 [a,b] | 10.9±0.17 [b] | 2.15±0.09 [b] | 230.82±2.3 [a,b] | 121±2.8 [a,b] |
| HFD/STZ | 348±5.5 [a] | 35.12±0.35 [a] | 30.17±0.21 [a] | 44.36±0.95 [a] | 274±3.1 [a] |
| Se-NPs (0.1mg) Trx | 110±3.4 [a,b] | 15.63±0.23 [a,b] | 4.24±0.14 [a,b] | 119.71±1.03 [a,b] | 330±3.4 [a,b] |
| Se-NPs (0.4mg) Trx | 108±3.3 [a,b] | 14.84±0.16 [a,b] | 3.95±0.17 [a,b] | 118.72±1.05 [a,b] | 320±3.6 [a,b] |
| MET Trx | 102±2.7 [b] | 13.84±0.19 [a,b] | 3.48±0.089 [a,b] | 127.75±1.1 [b] | 300±3.1 [a,b] |
| Se-NPs (0.1mg)-MET Trx | 98±2.9 [b] | 13.63±0.24 [a,b] | 3.29±0.11 [b] | 140.19±1.2 [b] | 307±3.3 [a,b] |
| Se-NPs (0.4mg)-MET Trx | 90±2.8 [b] | 11.89±0.22 [b] | 2.64±0.13 [b] | 158.53±1.1 [a,b] | 315±2.9 [a,b] |

Values are mean ± SE (n = 10). Significant changes (p<0.01) with respect to control and HFD/STZ-experimental rats are expressed by the letters (a) and (b), respectively.

## Fasting blood glucose level

HFD/STZ-induced T2DM resulted in a significant increase (P < 0.01) in fasting blood glucose levels (387±5.5 mg/dl) compared with control rats. The diabetic rats treated with two doses Se-NPs (0.1 or 0.4 mg/kg) a monotherapy for 8 weeks showed a significant reduction (P < 0.01) in the fasting blood glucose level that was noted compared to the untreated diabetic rats (3 fold decrease) (Table 2). Further, the treatment with standard anti-diabetic drug MET (100 mg/kg) was linked with a significant (P < 0.01) improvement in fasting blood glucose levels in diabetic rats. Moreover, the strategy of combined therapy in the treatment of diabetic rats (0.1 mg/kg Se-NPs-MET) and/or (0.4 mg/kg Se-NPs-MET) were displayed more efficient anti-hyperglycemic activity than Se-NPs or MET alone with maximum decline (P < 0.01) in blood glucose levels compared to untreated rats with levels near to the control values.

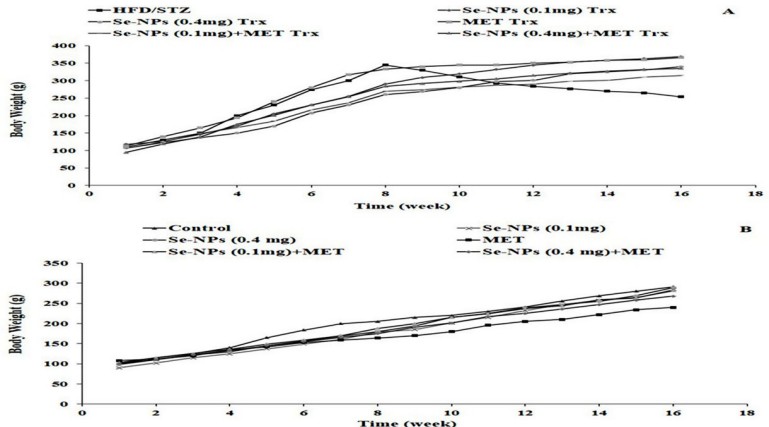

**Fig 4. Changes in body weights in induced and control groups after HFD/STZ induction and treatments.** A: Body weights of HFD/STZ-induced rats and after Se-NPs and MET treatment. B: Body weights of control groups that take a normal diet.

**Table 3. The comparison of the effect of Se-NPs, MET monotherapy, and combined therapy on serum liver functions ALT, AST, ALP, GGT, total bilirubin, albumin, and total protein of the studied groups.**

| Groups | ALT (U/l) | AST (U/l) | GGT (U/l) | Total Bilirubin (mg/dl) | ALP (U/l) | Total protein (g/dl) | Albumin (g/dl) |
|---|---|---|---|---|---|---|---|
| Control | 29±0.88 [b] | 35±1.02 [b] | 8.9±0.99 [b] | 0.42±0.09 [b] | 31±1.5 [b] | 6.14±0.117 [b] | 3.1±0.099 [b] |
| Se-NPs (0.1mg) | 25±1.1 [b] | 31±1.2 [b] | 8.8±1.23 [b] | 0.3±0.04 [b] | 42±2.1 [a,b] | 6.03±0.125 [b] | 2.9±0.1 [b] |
| Se-NPs (0.4 mg) | 22±1.5 [b] | 30±0.99 [b] | 8.2±0.89 [b] | 0.3±0.02 [b] | 40±1.3 [b] | 6.1±0.13 [b] | 3.01±0.08 [b] |
| MET | 23±1.2 [b] | 33±0.98 [b] | 7.9±1.5 [b] | 0.316±0.05 [b] | 40±1.45 [b] | 6.2±0.14 [b] | 3±0.093 [b] |
| Se-NPs (0.1mg)-MET | 21±1.08 [b] | 28±1.03 [b] | 7.7±0.88 [b] | 0.39±0.04 [b] | 35±1.8 [b] | 6.28±0.134 [b] | 3.6±0.08 [a,b] |
| Se-NPs (0.4 mg)-MET | 20±1.7 [b] | 27±1.04 [a,b] | 6.9±0.77 [a,b] | 0.32±0.02 [b] | 33±1.2 [b] | 6.5±0.17 [b] | 3.7±0.07 [a,b] |
| HFD/STZ | 121±2.9 [a] | 169±0.99 [a] | 25.3±0.78 [a] | 1.9±0.09 [a] | 190±2.4 [a] | 3.2±0.18 [a] | 1.9±0.11 [a] |
| Se-NPs (0.1mg) Trx | 37±0.59 [a,b] | 40±1.1 [b] | 11.3±1.14 [a,b] | 0.68±0.1 [a,b] | 48±2.2 [a,b] | 5.78±0.09 [b] | 2.5±0.098 [b] |
| Se-NPs (0.4mg) Trx | 35±0.99 [a,b] | 42±0.89 [a,b] | 10.7±1.06 [b] | 0.6±0.05 [a,b] | 45±1.09 [a,b] | 5.8±0.16 [b] | 2.7±0.1 [b] |
| MET Trx | 39±1.04 [a,b] | 42±0.78 [b] | 12.4±1.04 [a,b] | 0.6±0.06 [a,b] | 46±1.56 [a,b] | 5.73±0.08 [b] | 2.6±0.07 [a,b] |
| Se-NPs (0.1mg)-MET Trx | 33±0.85 [b] | 35±0.94 [b] | 10.2±1.1 [b] | 0.39±0.03 [b] | 35±1.9 [b] | 5.89±0.11 [b] | 2.9±0.102 [b] |
| Se-NPs (0.4mg)-MET Trx | 29±0.89 [b] | 33±1.22 [b] | 9.2±1.04 [b] | 0.3±0.02 [b] | 30±1.4 [b] | 6.1±0.13 [b] | 3.1±0.12 [b] |

Values are mean ± SE (n = 10). Significant changes (p<0.01) with respect to control and HFD/STZ-experimental rats are expressed by the letters (a) and (b), respectively.

## Serum insulin levels

Table 2 demonstrates the effect of Se-NPs (0.1 or 0.4 mg/kg) and/or MET alone or the combined nano-therapy and MET on the level of fasting serum insulin, the extent of HOMA-IR, and the function of pancreatic β-cell in HFD/STZ-induced rats. Our results showed that eight weeks of mono-therapeutic Se-NPs (0.1 or 0.4 mg/kg) and/or MET alleviate T2DM-induced elevation in the levels of insulin (3-fold) and HOMA-IR with a reduction in beta-cell function (HOMA-β) in rats. Furthermore, a combination of two doses Se-NPs (0.1 or 0.4 mg/kg) with MET displayed a more significant anti-IR, anti-hyperinsulinemic and anti-beta cell dysfunction activity that declines the insulin and HOMA-IR with increasing beta cell function significantly (p < 0.01) compared to untreated rats.

## Serum liver and kidney function parameters

HFD/STZ-induced rats showed elevated activities of hepatic enzymes (ALT, AST, GGT and ALP) with a reduction in hepatic functional protein levels (albumin and total protein) (Table 3) as well as an increment of renal functional-specific markers (creatinine, uric acid and urea) (Table 4) compared to control rats. These markers were significantly declined (p < 0.01) in the HFD/STZ rats treated with mono-therapeutic-two doses of Se-NP (0.1 or 0.4 mg/kg) and a similar effect was shown by MET treatment. The combined therapy of Se-NP (0.1 or 0.4 mg/kg) and MET was returned the elevated hepatic and renal markers close to normal levels compared to untreated rats. Serum levels of hepatic functional markers (albumin and total protein) showed a significant elevation after treatment with two doses of Se-NP (0.1 or 0.4 mg/kg) and a similar effect was shown by MET treatment. Moreover, combined therapy elevates the levels of (albumin and total protein) significantly (p < 0.01) near to normal values.

## Lipid profile components

HFD/STZ-induced T2D in rats resulted in a significant elevation (P < 0.01) in serum total lipid, TG, TC, LDL-c, and reduction in HDL-c levels compared to control rats. As shown in (Table 5), the diabetic rats that treated with two doses of Se-NP (0.1 or 0.4 mg/kg) and/or

**Table 4. The comparison of the effect of Se-NPs, MET monotherapy, and combined therapy on serum kidney functions uric acid, urea, and creatinine of the studied groups.**

| Groups | Creatinine (mg/dl) | Urea (mg/dl) | Uric acid (mg/dl) |
|---|---|---|---|
| Control | 0.751±0.017 [b] | 20.6±0.8 [b] | 4.14±0.27 [b] |
| Se-NPs (0.1mg) | 0.785±0.04 [b] | 22.8±0.53 [b] | 4.29±0.21 [b] |
| Se-NPs (0.4 mg) | 0.71±0.03 [b] | 20.4±0.44 [b] | 4.1±0.13 [b] |
| MET | 0.82±0.03 [a,b] | 23.4±0.89 [b] | 4.8±0.15 [a,b] |
| Se-NPs (0.1mg)-MET | 0.684±0.07 [a,b] | 17.8±0.75 [b] | 4.13±0.29 [b] |
| Se-NPs (0.4 mg)-MET | 0.56±0.05 [a,b] | 16.01±0.54 [a,b] | 3.8±0.25 [b] |
| HFD/STZ | 2.5±0.02 [a] | 66.7±1.1 [a] | 13.5±0.33 [a] |
| Se-NPs (0.1mg) Trx | 0.95±0.07 [a,b] | 31.14±0.8 [a,b] | 5.84±0.44 [a,b] |
| Se-NPs (0.4mg) Trx | 0.85±0.01 [a,b] | 30.01±0.34 [a,b] | 5.6±0.34 [a,b] |
| MET Trx | 1.2±0.03 [a,b] | 30.32±0.52 [a,b] | 5.54±0.23 [a,b] |
| Se-NPs (0.1mg)-MET Trx | 0.89±0.08 [a,b] | 28.6±0.79 [a,b] | 5.09±0.12 [a,b] |
| Se-NPs (0.4mg)-MET Trx | 0.71±0.05 [b] | 27.4±0.56 [a,b] | 4.4±0.16 [b] |

Values are mean ± SE (n = 10). Significant changes (p<0.01) with respect to control and HFD/STZ-experimental rats are expressed by the letters (a) and (b), respectively.

MET showed an anti-hyperlipidemic activity with a significant decrease (P < 0.01) in lipid profile compared to untreated rats with the exception of serum HDL-c level. Treatment with two doses Se-NP (0.1 or 0.4 mg/kg) combined with standard drug MET showed a more potent effect than monotherapy in decreasing dyslipidemia in T2D rats by a significant decline in serum total lipid, TG, TC, and LDL-c with a marked elevation of HDL-c level compared with the diabetic-untreated group.

## AGEs levels in type 2 diabetic rats

T2DM and chronic hyperglycemia considered a major cause of AGEs production and accumulation in blood and tissues resulting in diabetic complications. As exhibited in (Fig 5),

**Table 5. The comparison of the effect of Se-NPs, MET monotherapy, and combined therapy on serum Lipid profile TC, TG, LDL-c, HDL-c, TG, total lipid of the studied groups.**

| Groups | Cholesterol | TG | LDL-c | HDL-c | Total Lipid |
|---|---|---|---|---|---|
| Control | 102±2.6 [b] | 83±1.4 [b] | 38.2±1.1 [b] | 42±1.7 [b] | 431±7.2 [b] |
| Se-NPs (0.1mg) | 95±2.55 [b] | 85±2.1 [b] | 35.4±1.6 [b] | 45±0.85 [b] | 405±8.4 [a,b] |
| Se-NPs (0.4 mg) | 90±2.2 [b] | 83±2.3 [b] | 34±1.4 [b] | 45±1.3 [b] | 402±8.6 [a,b] |
| MET | 87±2.09 [a,b] | 77±1.33 [b] | 27.8±0.9 [a,b] | 48±1.5 [b] | 383±7.99 [a,b] |
| Se-NPs (0.1mg)-MET | 72±1.3 [a,b] | 71±1.04 [a,b] | 22.2±0.89 [a,b] | 51±1.88 [a,b] | 365±5.7 [a,b] |
| Se-NPs (0.4 mg)-MET | 70±1.99 [a,b] | 64±1.89 [a,b] | 20.4±1.5 [a,b] | 56±1.3 [a,b] | 355±7.5 [a,b] |
| HFD/STZ | 299±4.1 [a] | 278±2.9 [a] | 198±2.4 [a] | 14±1.3 [a] | 1750±10.3 [a] |
| Se-NPs (0.1mg) Trx | 103±2.6 [b] | 108±1.78 [a,b] | 41±1.1 [b] | 34±1.45 [a,b] | 521±6.33 [a,b] |
| Se-NPs (0.4mg) Trx | 100±1.3 [b] | 103±1.5 [a,b] | 39.2±1.1 [b] | 35±1.7 [b] | 520±8.5 [a,b] |
| MET Trx | 99±1.9 [b] | 97±1.05 [a,b] | 33.4±1.2 [b] | 37±0.99 [b] | 471±9.1 [a,b] |
| Se-NPs (0.1mg)-MET Trx | 89±2.3 [a,b] | 87±1.9 [b] | 29.1±1.24 [a,b] | 40±1.4 [b] | 447±7.9 [a,b] |
| Se-NPs (0.4mg)-MET Trx | 85±1.6 [a,b] | 80±1.8 [b] | 27.09±1.5 [a,b] | 49±1.7 [b] | 438±8.5 [b] |

All units are expressed as mg/dl and the mean values ± SE (n = 10). Significant changes (p<0.01) with respect to control and HFD/STZ-experimental rats are expressed by the letters (a) and (b), respectively.

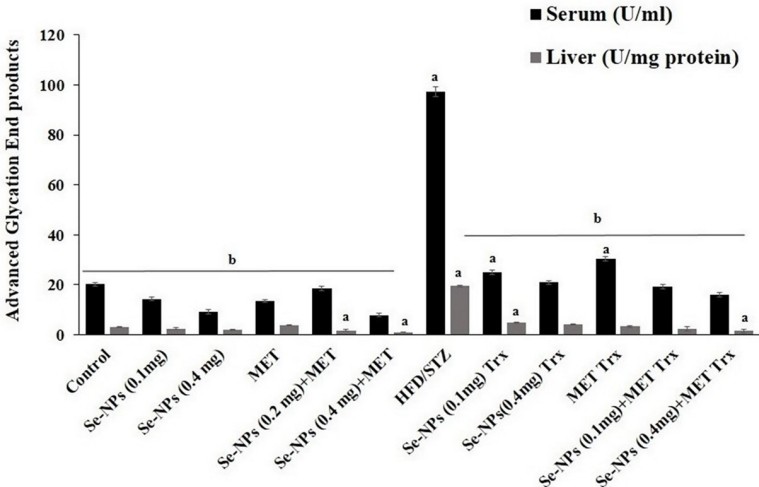

**Fig 5. Effect of Se-NPs and MET on AGEs levels of serum and liver tissues in HFD/STZ-induced rats.** Values are mean ± SE ($n = 10$). Significant changes ($p<0.01$) with respect to control and HFD/STZ-experimental rats are expressed by the letters (a) and (b), respectively.

serum and liver AGEs levels in HFD/STZ-induced rats were significantly increased (4.8, 5.9 fold, respectively) when compared to untreated rats. Monotherapy plane of two doses of Se-NP (0.1 or 0.4 mg/kg) showed a significant decrease ($p < 0.01$) in serum and hepatic AGEs levels, MET also decreased AGEs level significantly ($p < 0.01$). Interestingly, the combined therapy strategy displayed a more reduction ($p < 0.01$) in AGEs levels in serum and hepatic rats close to control values compared to diabetic untreated rats as well as monotherapy.

## Effect of Se-NPs on oxidant and antioxidant levels

To study the effect of T2DM with respect to hepatic oxidative damage, we assessed the action of the antioxidant capacity by measuring the enzymatic and non-enzymatic antioxidants. To investigate whether monotherapy of two doses Se-NPs (0.1 or 0.4 mg/kg) and standard anti-diabetic drug MET affect oxidative stress and boost antioxidant status in T2DM rats. We measured MDA, NO and XO oxidative stress markers with an evaluation of enzymatic antioxidants (total-SOD, CAT, GPx, GR and GST) as well as non-enzymatic levels of (GSH, TAC, and AHR) in the serum and hepatic tissues of rats. Current data showed that HFD/STZ significantly increased the serum and hepatic MDA, NO and XO levels compared with that of the control group (Table 6). The two doses (0.1 or 0.4 mg/kg) Se-NPs-treated groups significantly reduced serum and hepatic MDA, NO and XO levels compared to untreated rats. Anti-diabetic drug MET showed a significant reduction in serum and hepatic MDA, NO and XO levels compared to untreated rats with a lesser effect than Se-NPs treatment as a monotherapy. Over and above, enzymatic (total-SOD, CAT, GPx, GR and GST) (Table 7) and non-enzymatic (GSH, TAC, and AHR) (Table 8) levels of antioxidants demonstrated a significant reduction in HFD/STZ-induced rats compared to control group. Monotherapy treatment strategy of two doses (0.1 or 0.4 mg/kg) Se-NPs and anti-diabetic drug MET attenuated the HFD/STZ-induced reduction and showed a significant elevation in serum and hepatic levels of enzymatic and non-enzymatic antioxidants compared to untreated rats. Regarding combined therapy strategy, especially (0.4 mg/kg Se-NPs-MET) revealed a more efficient elevation in enzymatic and non-enzymatic levels of serum and hepatic tissues compared to untreated groups and also compared to monotherapy linked with a marked decline in oxidative stress markers (MDA,

**Table 6. Changes in oxidative stress markers [MDA, NO, and XO] in serum and liver tissue of HFD/STZ-induced rats and after treatment with Se-NPs, MET monotherapy, and combined therapy.**

| Groups | | MDA (nmol/mg protein) | NO (μM/mg protein) | XO (μmol/h/mg protein) |
|---|---|---|---|---|
| Control | Liver | 51.28±3.24 [b] | 2.3±0.14 [b] | 15.34±0.44 [b] |
| | Serum | 33.3±2.33 [b] | 2.98±0.29 [b] | 10.3±0.33 [b] |
| Se-NPs (0.1mg) | Liver | 45.3±4.43 [b] | 1.81±0.15 [b] | 12.4±0.35 [b] |
| | Serum | 23.8±2.85 [a,b] | 1.01±0.12 [a,b] | 9.09±0.35 [b] |
| SeNPs (0.4 mg) | Liver | 41.4±3.2 [a,b] | 1.5±0.3 [a,b] | 10.2±0.23 [a,b] |
| | Serum | 21.3±2.2 [a,b] | 1.00±0.11 [a,b] | 8.2±0.14 [b] |
| MET | Liver | 40.5±2.12 [a,b] | 2.1±0.09 [b] | 14.3±0.12 [b] |
| | Serum | 22.5±3.79 [a,b] | 1.24±0.12 [a,b] | 11.3±0.49 [b] |
| Se-NPs (0.1mg)-MET | Liver | 34.9±3.35 [a,b] | 1.07±0.23 [a,b] | 13.3±0.45 [b] |
| | Serum | 19.9±3.98 [a,b] | 0.94±0.13 [a,b] | 8.01±0.38 [b] |
| Se-NPs (0.4 mg)-MET | Liver | 28.4±3.3 [a,b] | 0.89±0.35 [a,b] | 8.03±0.11 [a,b] |
| | Serum | 15.4±2.5 [a,b] | 0.84±0.09 [a,b] | 6.9±0.23 [a,b] |
| HFD/STZ | Liver | 891.1±8.33 [a] | 19.3±0.58 [a] | 110.23±0.96 [a] |
| | Serum | 211.9±5.55 [a] | 10.16±0.51 [a] | 45.5±0.55 [a] |
| Se-NPs (0.1mg) Trx | Liver | 122.3±4.08 [a,b] | 7.01±0.12 [a,b] | 27.8±0.88 [a,b] |
| | Serum | 39.37±4.16 [b] | 3.79±0.05 [a,b] | 20.4±0.16 [a,b] |
| Se-NPs (0.4mg) Trx | Liver | 118.5±2.1 [a,b] | 6.3±0.12 [a,b] | 23.1±0.2 [a,b] |
| | Serum | 32.1±1.9 [b] | 3.01±0.08 [b] | 18.04±0.12 [a,b] |
| MET Trx | Liver | 235.2±5.14 [a,b] | 5.4±0.23 [a,b] | 25.4±0.74 [a,b] |
| | Serum | 50.3±3.94 [a,b] | 4.06±0.07 [a,b] | 25.5±0.94 [a,b] |
| Se-NPs (0.1mg)-MET Trx | Liver | 104.3±5.78 [a,b] | 3.2±0.09 [a,b] | 20.44±0.47 [a,b] |
| | Serum | 30.2±4.8 [b] | 2.89±0.14 [b] | 17.3±0.43 [a,b] |
| Se-NPs (0.4mg)-MET Trx | Liver | 95.4±5.2 [a,b] | 2.1±0.11 [b] | 16.04±0.67 [b] |
| | Serum | 26.03±5.1 [b] | 2.11±0.16 [b] | 14.94±0.63 [b] |

Significant changes ($p < 0.01$) with respect to control and HFD/STZ-experimental rats are expressed by the letters (a) and (b), respectively.

NO, and XO). Thus, Se-NP imitated to help retrieve the impaired activity of enzymatic and non-enzymatic antioxidants in T2D-induced rats.

## Inhibitory effect of Se-NPs on the levels of inflammatory mediators in HFD/STZ-induced rats

Two doses of Se-NPs affect the serum and hepatic protein levels of inflammatory markers in HFD/STZ-induced rats. The anti-inflammatory effects of two doses Se-NPs and MET were investigated by measuring the serum and hepatic inflammatory cytokines. Compared to control rats, the levels of inflammatory cytokines in serum and liver, including IL-6, IL-1β, TNF-α, and iNOS were increased significantly in HFD/STZ-induced rats. The hyper-inflammation effect was mitigated by two doses of Se-NPs (0.1 or 0.4 mg/kg) administration, and the same results were shown in MET treatment compared to untreated rats. Moreover, the anti-inflammatory effect of the combined therapy displayed more potent in a reduction of serum and hepatic inflammatory cytokines (iNOS, TNF-α, IL-6, and IL-1β) compared to untreated rats, particularly the (0.4 mg/kg Se-NPs-MET), which had a more reducing effect than other treatment strategies (Fig 6A, 6B, 6C, and 6D). In addition, we investigated the effects of two doses Se-NPs (0.1 or 0.4 mg/kg) and MET on hepatic protein levels of western blot analysis of the inflammatory cytokines p65-NF-κB and COX-2. HFD/STZ-induced rats remarkably increased

**Table 7. Evaluation of the antioxidant enzyme variations (Total-SOD, CAT, GPx, GR, and GST) in serum and liver tissue of HFD/STZ-induced rats and after treatment with Se-NPs, MET monotherapy, and combined therapy.**

| Groups | | CAT activity (µg/min/mg protein) | GPx (µmol/min/mg protein) | GR (µmol/min/mg protein) | GST (µmol/min/mg protein) | Total SOD (U/mg protein) |
|---|---|---|---|---|---|---|
| Control | Liver | 7.8±0.17 [b] | 56.7±1.07 [b] | 7.9±0.14 [b] | 1.9±0.04 [b] | 7.9±0.14 [b] |
| | Serum | 5.3±0.23 [b] | 30.04±2.25 [b] | 5.5±0.13 [b] | 1.7±0.03 [b] | 4.01±0.13 [b] |
| Se-NPs (0.1mg) | Liver | 11.9±0.09 [a,b] | 62.8±1.54 [a,b] | 9.3±0.09 [b] | 2.4±0.09 [b] | 10.2±0.19 [a,b] |
| | Serum | 7.1±0.25 [b] | 35.5±2.25 [b] | 6.66±0.25 [b] | 2.01±0.05 [b] | 4.9±0.15 [b] |
| Se-NPs (0.4 mg) | Liver | 13.8±0.12 [a,b] | 65.7±1.54 [a,b] | 10.0±0.12 [a,b] | 2.6±0.08 [b] | 11.2±0.09 [a,b] |
| | Serum | 8.3±.25 [a,b] | 41.2±2.05 [a,b] | 7.4±0.11 [a,b] | 2.2±0.06 [b] | 5.3±0.17 [a,b] |
| MET | Liver | 10.03±0.12 [a,b] | 59.03±1.12 [b] | 8.2±0.12 [b] | 2.25±0.05 [b] | 8.8±0.1 [b] |
| | Serum | 4.9±0.19 [b] | 29.03±2.19 [b] | 5.98±0.29 [b] | 1.8±0.09 [b] | 4.4±0.19 [b] |
| Se-NPs (0.1mg)-MET | Liver | 17.01±0.13 [a,b] | 68.3±2.01 [a,b] | 11.9±0.15 [a,b] | 2.75±0.03 [a,b] | 11.9±0.13 [a,b] |
| | Serum | 9.1±0.22 [a,b] | 43.09±1.98 [a,b] | 7.9±0.18 [a,b] | 2.34±0.08 [a,b] | 5.3±0.18 [a,b] |
| Se-NPs (0.4 mg)-MET | Liver | 22.3±0.23 [a,b] | 73.9±2.13 [a,b] | 15.3±0.11 [a,b] | 3.4±0.09 [a,b] | 12.6±0.08 [a,b] |
| | Serum | 12.3±0.25 [a,b] | 48.3±1.90 [a,b] | 9.5±0.13 [a,b] | 2.6±0.07 [a,b] | 5.9±0.12 [a,b] |
| HFD/STZ | Liver | 2.2±0.08 [a] | 13.9±1.08 [a] | 1.5±0.06 [a] | 0.15±0.04 [a] | 0.16±0.03 [a] |
| | Serum | 1.2±0.11 [a] | 9.9±1.31 [a] | 0.77±0.15 [a] | 0.1±0.05 [a] | 0.56±0.02 [a] |
| Se-NPs (0.1mg) Trx | Liver | 8.3±0.21 [b] | 48.04±1.21 [a,b] | 7.1±0.08 [b] | 1.7±0.08 [b] | 7.7±0.18 [b] |
| | Serum | 6.7±0.18 [b] | 33.3±0.88 [b] | 5.2±0.16 [b] | 1.5±0.06 [b] | 3.99±0.16 [b] |
| Se-NPs (0.4mg) Trx | Liver | 10.2±0.13 [a,b] | 52.9±1.01 [b] | 8.9±0.1 [b] | 1.9±0.07 [b] | 8.4±0.08 [b] |
| | Serum | 7.2±0.17 [a,b] | 39.1±1.18 [a,b] | 6.3±0.14 [b] | 1.62±0.04 [b] | 4.5±0.14 [b] |
| MET Trx | Liver | 7.3±0.14 [b] | 44.4±1.49 [a,b] | 6.8±0.14 [b] | 1.35±0.06 [b] | 6.89±0.12 [b] |
| | Serum | 4.1±0.12 [b] | 23.9±1.24 [a,b] | 4.6±0.24 [b] | 1.1±0.09 [b] | 3.03±0.19 [b] |
| Se-NPs (0.1mg)-MET Trx | Liver | 11.4±0.11 [a,b] | 59.3±2.03 [b] | 10.4±0.27 [a,b] | 2.1±0.07 [b] | 8.6±0.19 [b] |
| | Serum | 9.3±0.31 [a,b] | 41.2±1.83 [a,b] | 6.9±0.26 [b] | 1.79±0.03 [b] | 4.9±0.16 [b] |
| Se-NPs (0.4mg)-MET Trx | Liver | 17.4±0.14 [a,b] | 64.3±2.13 [a,b] | 12.3±0.29 [a,b] | 2.6±0.09 [b] | 9.4±0.13 [a,b] |
| | Serum | 11.1±0.33 [a,b] | 45.2±1.86 [a,b] | 7.5±0.22 [a,b] | 1.98±0.05 [b] | 5.2±0.13 [a,b] |

Significant changes ($p<0.01$) with respect to control and HFD/STZ-experimental rats are expressed by the letters (a) and (b), respectively.

protein levels of p65-NF-κB, and COX-2 compared to the control group, whereas these effects were attenuated by monotherapy of two doses of Se-NPs (0.1 or 0.4 mg/kg) and MET. Furthermore, combined therapy of two doses of Se-NPs (0.1 or 0.4 mg/kg) and MET had a more potent effect on reducing the protein levels of p65-NF-κB and COX-2 (Fig 6E and 6F) compared to untreated rats and monotherapy.

## Hepatic inflammatory mediators Gene expression

Pro-inflammatory mediator's augmentation revealed to play causative roles in T2DM-induced IR. Clinical attempts targeting single pro-inflammatory markers in order to treat IR. The current study was carried to determine how Se-NPs and MET treatment as a monotherapy and combined therapy lower hepatic inflammatory and oxidative mediator levels by altering the gene expression of pro-inflammatory cytokines and chemokines; il-1α, il-23, mcp-1, socs-3, Ox-LDL-1. In hepatic tissue, the overexpression of pro-inflammatory cytokines and chemokines genes were shown in HFD/STZ-induced rats than that in the overall control rats. In contrast, the expression pro-inflammatory cytokines and chemokines genes were repressed by the anti-inflammatory and anti-oxidative effect of two doses of Se-NPs (0.1 or 0.4 mg/kg), similarly in MET treated group compared to untreated rats. Interestingly, a combination of Se-NPs (0.1 or 0.4 mg/kg) with standard drug MET revealing a more efficient anti-inflammatory

**Table 8. Alteration of the level of non-enzymatic antioxidant proteins [GSH, AHR, and TAC] in serum and liver tissue of HFD/STZ-induced rats and after treatment with Se-NPs, MET monotherapy, and combined therapy.**

| Groups | | GSH (µg/mg protein) | AHR (U/mg protein) | TAC (%) |
|---|---|---|---|---|
| **Control** | Liver | 101.11±1.24 [b] | 292.2±3.24 [b] | 77±1.43[b] |
| | Serum | 35.2±1.03 [b] | 75.7±2.33 [b] | 33±0.33 [b] |
| **Se-NPs (0.1mg)** | Liver | 129.5±1.59 [a,b] | 303.3±4.09 [b] | 78±1.56 [b] |
| | Serum | 41.3±1.05 [b] | 79.3±2.85 [b] | 39±0.85 [b] |
| **Se-NPs (0.4 mg)** | Liver | 134.2±2.1 [a,b] | 310.2±3.6 [a,b] | 80±1.23 [b] |
| | Serum | 44.5±1.3 [a,b] | 81.2±3.1 [b] | 41±0.88 [a,b] |
| **MET** | Liver | 126.66±1.11 [a,b] | 298.2±2.12 [b] | 66±1.12 [a,b] |
| | Serum | 38.4±1.19 [b] | 71.2±2.79 [b] | 32±0.79 [b] |
| **Se-NPs (0.1mg)-MET** | Liver | 154.3±2.03 [a,b] | 320.3±3.35 [a,b] | 85±1.3 [b] |
| | Serum | 45.6±1.88 [a,b] | 85.3±2.98 [a,b] | 44±0.45 [a,b] |
| **Se-NPs (0.4 mg)-MET** | Liver | 159.3±1.6 [a,b] | 333.4±3.4 [a,b] | 89±1.44 [a,b] |
| | Serum | 49.2±1.4 [a,b] | 92.1±3.4 [a,b] | 49±0.98 [a,b] |
| **HFD/STZ** | Liver | 34.92±0.55 [a] | 72.22±2.26 [a] | 23±0.98 [a] |
| | Serum | 10.1±0.29 [a] | 12.4±2.55 [a] | 12±0.08 [a] |
| **Se-NPs (0.1mg) Trx** | Liver | 87.45±1.88 [a,b] | 270.9±4.08 [a,b] | 60±1.56 [a,b] |
| | Serum | 29.8±2.16 [b] | 67.6±3.16 [b] | 31±0.16 [b] |
| **Se-NPs (0.4mg) Trx** | Liver | 90.4±1.2 [b] | 283.2±4.1 [b] | 64±1.2 [a,b] |
| | Serum | 32.2±1.6 [b] | 69.3±3.3 [b] | 35±0.23 [b] |
| **MET Trx** | Liver | 66.43±2.12 [a,b] | 232.3±3.14 [a,b] | 53±1.73 [a,b] |
| | Serum | 30.3±2.19 [b] | 59.9±2.94 [a,b] | 29±0.94 [b] |
| **Se-NPs (0.1mg)-MET Trx** | Liver | 98.1±2.09 [b] | 289.3±3.47 [b] | 73±1.55 [b] |
| | Serum | 33.4±1.34 [b] | 70.8±3.83 [b] | 37±0.83 [a,b] |
| **Se-NPs (0.4mg)-MET Trx** | Liver | 107.3±2.02 [b] | 301.1±3.47 [b] | 78±1.55 [b] |
| | Serum | 36.4±1.44 [b] | 75.3±3.83 [b] | 40±0.83 [a,b] |

Significant changes (p<0.01) with respect to control and HFD/STZ-experimental rats are expressed by the letters (a) and (b), respectively.

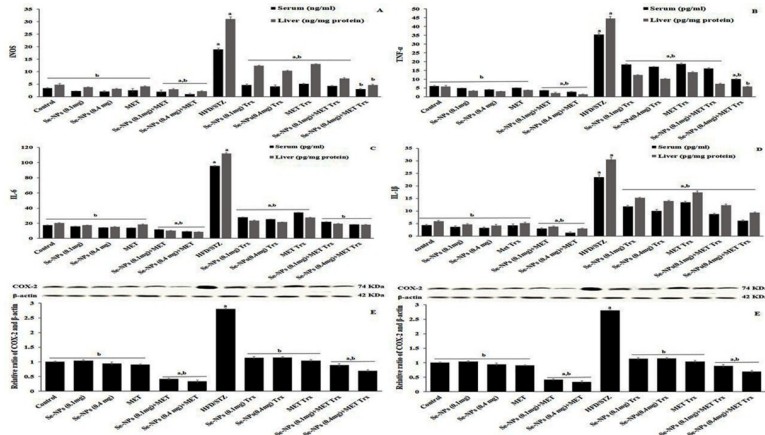

**Fig 6. A graphical presentation of serum and hepatic inflammatory mediators in HFD/STZ-induced rats treated with Se-NPs, MET monotherapy, and combined therapy.** Levels of inflammatory cytokines, including iNOS (A), TNF-α (B), IL-6 (C) and IL-1β (D), were measured in serum and liver tissues by ELISA (n = 5). Protein expression levels of COX-2 (E) and p-p65 (F), were measured in liver tissues by western blot, β-Actin was used as a loading control. Values are expressed as mean ± SE (the mean of three assays for western blot). Significant changes (p<0.01) with respect to control and HFD/STZ-experimental rats are expressed by the letters (a) and (b), respectively.

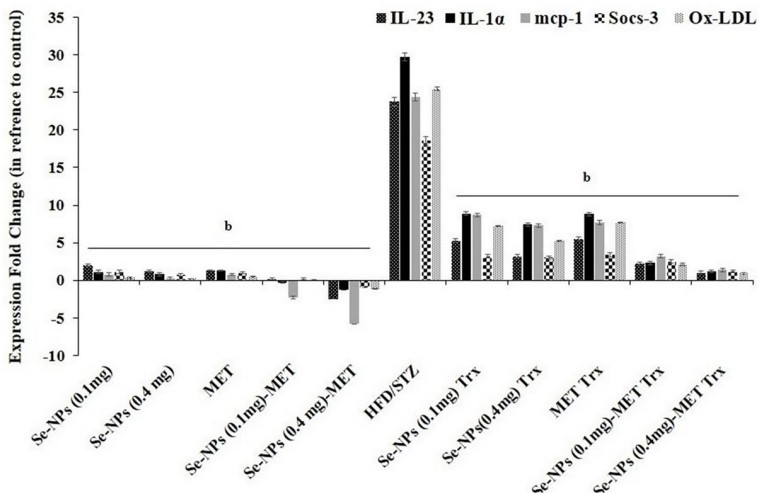

**Fig 7. Effect of Se-NPs, MET monotherapy, and combined therapy on hepatic mRNA expression of inflammatory genes in control and HFD/STZ-experimental groups.** Values are expressed as fold induction ± SE (the mean of three assays) relative to the mRNA level in the control and were normalized to β-actin RNA level. Significant changes (p<0.01) with respect to HFD/STZ-experimental rats are expressed by the letter b.

action displayed by significant repression of gene expression of pro-inflammatory cytokines and chemokines (il-1α, il-23, mcp-1, socs-3, Ox-LDL-1) compared to untreated rats and values of monotherapy groups (Fig 7).

## Se-NPs ameliorated liver insulin signaling sensitivity

To investigate the molecular mechanisms underlying the anti-diabetic effect of Se-NPs, we studied the insulin signaling pathway in the liver of HFD/STZ-induced rats, which plays a pivotal role in glucose homeostasis. The rats were estimated for the presence of pIRS-1, pAKT, pGSK-3β, and pAMPK. As displayed in (Fig 8A, 8B, 8C and 8D), the hepatic protein expression levels

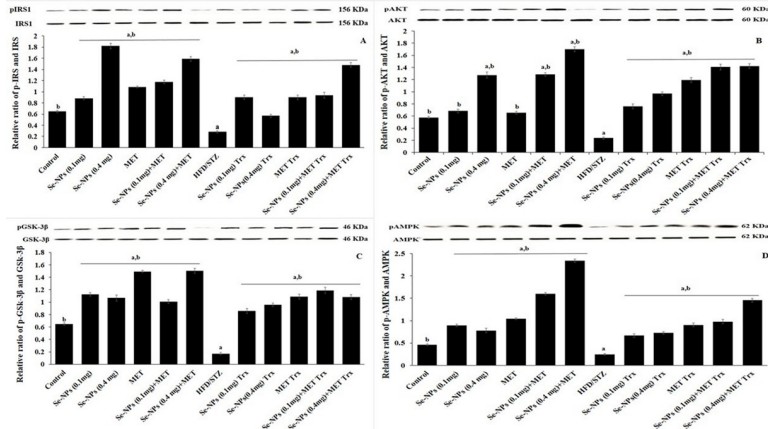

**Fig 8. Western blot analyses show the expression levels of insulin signaling pathway.** Liver anti IRS1, anti-IRS1 p$^{Tyr612}$, anti-AKT1 & anti-AKT1 p$^{ser473}$, anti-GSK-3β & anti-GSK-3β p$^{Ser9}$, anti-AMPK-α, anti-AMPK- α p$^{T172}$, protein levels were measured by western blot. The quantitative values were normalized to IRS1, AKT, GSK-3β, and AMPK-α, respectively in liver tissue of HFD/STZ-induced rats and after treatment with Se-NPs and MET monotherapy and combined therapy. Values are mean ± SE (the mean of three assays). Significant changes (p<0.01) with respect to control and HFD/STZ-experimental rats are expressed by the letters (a) and (b), respectively.

of pIRS-1, pAKT pGSK-3β, and pAMPK were diminished significantly in HFD/STZ-induced IR in rats compared to normal control. The attenuation in insulin signaling pathway was alleviated after administration of two doses Se-NPs (0.1 or 0.4 mg/kg) as well as anti-diabetic medication, MET by a remarkable increase in levels of phosphorylated signaling proteins (pIRS-1, pAKT, pGSK-3β, and pAMPK) in the liver compared to untreated rats. Interestingly, the up-regulation and improvement in the expression level of hepatic insulin signaling proteins were significantly increased in rats treated with combined therapy of two doses Se-NPs (0.1 or 0.4 mg/kg) and MET compared to untreated rats as well as mono-therapeutic strategy.

## Discussion

In our study, we used ascorbic acid as a reducing agent as it has good reducing properties and biocompatible and forming a spherical Se-NPs having a size range of 30 to 80 nm as measured by TEM. Dextrin was used for coating, which is biocompatible and inert. Se-NPs were screened for the cytotoxicity against rat hepatocytes cell line by MTT Assay. The lack of any noticeable cytotoxicity of synthesized Se-NPs supplies novel opportunities for the safe use in therapy. In addition, no detected hepatic and renal toxicity by Se-NPs in the tested animals as revealed by the non-signification alterations in hepatic and renal function parameters.

Excessive ROS generation causes damage to all tissues. The scavenging of ROS generated during the oxidation can be achieved by a strong antioxidant. In the present study, the scavenging values of Se-NPs increased with an increase in concentration. The Se-NPs manifested scavenging free radicals that increased with increasing concentrations of the Se-NPs. It reached a maximum scavenging power of DPPH, NO, hydrogen peroxide, reducing power and TAC of 81%, 84%, 77%, 83%, and 89%, respectively (Fig 3A, 3B, 3C, 3D and 3E). Due to their scavenging potency, antioxidants are used in the management of T2DM. Similarly, Lei Wang et al., [50] showed the antioxidant activity of biosynthesized Se-NPs.

In the current study, STZ was used to induce T2DM, which is an accepted model to explore the pathophysiology of T2DM. HFD/STZ protocol favors mimic-T2DM because the obesity precedes the development of T2DM. Interestingly, the final event contributing to the development of T2DM is a β-cell failure, so, STZ is used for induction T2DM in experimental animals [51]. Over and above, HFD/STZ administration demonstrated features of T2DM including hyperglycemia, hyperinsulinemia, and dyslipidemia similar to our previous study [52]. It has been reported that Se has features that mimic insulin. Further, MET is a standard anti-diabetic drug affecting blood glucose level through three ways: the deactivation of hepatic glucose production, restriction of glucose entrance from the intestine, and boosting the insulin function in the peripheral tissues [53]. MET was also found to enhance insulin secretion from beta cells [54]. The present study attempts to clarify the hypoglycemic effects of two doses Se-NPs (0.1 or 0.4 mg/kg) and/or MET as monotherapy and combined therapy in HFD/STZ-induced T2DM in rats. The HFD administration followed by injection of low dose STZ-induced insulin resistance, resulting in hyperglycemia, generation of ROS and hyperinsulinemia and thus T2DM [55]. The current study demonstrated that the T2DM-induced rats that were left untreated showed a remarkable elevation of blood glucose levels and insulin levels. In contrast, diabetic rats treated with monotherapy and combined therapy resulted in improved glycemic control and insulin levels with values closed to control rats [20]. This effect may be due to the ability of selenium to renew the activity of beta cells, stimulation of insulin release, and decreasing the blood glucose level [56].

It has been suggested that β-cell dysfunction is a major incident in the development of insulin resistance [57]. The current study displayed a significant β-cell dysfunction in HFD/STZ-induced rats by a decrease in the HOMA-β index. Se-NPs and MET attenuated β-cell

dysfunction in T2DM groups and this effect was markedly elevated in groups treated with combined therapy [58]. Further, the current study clearly confirmed the presence of IR in HFD/STZ-induced T2DM rats by increased HOMA-IR index similar to a previous study [59]. Administration of two doses Se-NPs and/or MET alone attenuated the HFD/STZ-induced insulin resistance in experimental animals. In addition, the combination of Se-NPs and MET exhibited more attenuation efficiency of IR in HFD/STZ-induced rats. These observations affirm the fact of the synergistic therapeutic effect of Se-NPs and MET against T2DM-induced hepatic insulin resistance and β-cell function improvement.

In fact, the liver is an essential organ for maintaining blood glucose levels, metabolic functions, storage, detoxification, and excretion. ALT and AST transaminases are closely linked biomarkers to hepatic tissue damage [60]. It has been established that the increase of serum activities of transaminases, GGT, and ALP in T2D-induced rats may be due to hepatic injury linked with STZ resulting in leakage of these enzymes from the liver into plasma [61]. The current study has shown that treatment with Se-NPs and/or MET as monotherapy and also combined therapy restored the activities of ALT, AST, GGT and ALP toward normal levels as well as the levels of albumin, total protein and bilirubin returned near to normal levels. These results could be due to the role of Se-NPs in protecting the integrity and functions of liver tissues besides its radical scavenging activity [20].

With regard to diabetic complications, much of the studies focusing on kidney functions, which involved in diabetic nephropathy development. In our study, serum levels of creatinine, urea, and uric acid were markedly elevated in untreated T2DM rats, which are clear signs of renal dysfunction [62]. The observed renal dysfunction in HFD/STZ rats was reversed by treatment with Se-NPs and/or MET monotherapy as well as combined-therapy. The hypoglycemic effect of Se-NPs and MET may explain the ameliorated renal function in the treated rats.

As demonstrated previously, T2DM is often accompanied by abnormalities in serum lipid profiles. Thus, T2D therapies are aimed to prevent dyslipidemia, which characterized by elevated serum levels of TG, TC, LDL-c, total lipids and decreased serum levels of HDL-c [22]. In the present study, a marked disorder in serum lipid profiles was observed in HFD/STZ-induced diabetic rats (Table 5). After the oral administration of monotherapy and combined-therapy of Se-NPs and MET a remarkable decrease in serum levels of TG, total lipid, TC, LDL-c were observed linked with a significant increase in HDL-c levels compared to the untreated diabetic rats. The decrease TG and TC levels in the current study may be due to the action of HDL-c that could raise the efflux of TG and TC to liver tissue for catabolism [63].

Moreover, comparing the combined therapy with monotherapy especially, the group treated with Se-NPs (0.4mg/kg) combined with MET in the levels of serum indices; we could conclude that the administration of this dose was more effective compared to other monotherapy or low dose combined therapy in attenuating the increase in these serum parameters. Therefore, the use of Se-NPs (0.4mg/kg) with MET in T2DM treatment may be a hopeful effective anti-diabetic supplement.

Many studies elucidate the implication of oxidative stress in T2DM pathogenesis and its complications in human and animal models as a common factor that leads up to the increasing of tissues-specific IR. Alongside, hyperglycemia in T2DM can markedly elevate the production of oxidative stress in tissues, and can also enhance the imbalance between the ROS production and the anti-oxidative protective system [64]. Thus, most therapies used in the treatment of T2DM focus on lowering blood glucose to a normal level. Moreover, HFD/STZ-induced T2DM via the generation of ROS, causing damage of β cells of the pancreas. So, it is essential to explore therapies against ROS production as standard treatments for T2DM.

It was reported that MET exhibits its antioxidant activities by suppressing mitochondrial respiration that increases the antioxidant enzyme activities and diminished the ROS in diabetic

rats as well as suppressing AGEs production via its hypoglycemic effect indirectly and through an insulin-dependent mechanism directly [23]. Moreover, recent studies demonstrated that dietary Se-NPs supplementation enhances the activity of GPx, total-SOD, and CAT in erythrocytes of layered chicks [65] than dietary selenium. This may be illustrated that nanoparticles exhibit high absorption efficiency due to their small size and large surface area [66]. Remarkably, the present results showed a reduction in serum and hepatic antioxidant enzyme activities (GST, GPx, total-SOD, CAT, and GR) and non-enzymatic antioxidants (GSH, TAC and AHR) with elevation of (MDA, NO and XO) levels that were noticed in HFD/STZ-induced diabetic rats, indicating the augmented oxidative stress in serum and liver tissue.

It is reasonable to deem that monotherapy and combined therapy showed a beneficial effect against the development of T2DM by exhibiting antioxidant activities in the T2DM rats. This efficiency was achieved by an increment of hepatic and serum antioxidant enzyme activities as well as non-enzymatic antioxidants linked with a remarkable decrease in oxidative stress markers in T2D rats. In addition, elevated blood glucose levels produce tissue damage by several mechanisms, involving AGEs formation in the blood and tissues. Our results also showed a significant elevation in serum and hepatic tissues of AGEs levels, and these elevations were recovered by to its normal values by the administration of monotherapy as well as a combined therapy. However, especially, the effect of combined Se-NPs (0.4 mg/kg BW) with MET on recovering of oxidant and antioxidant balance with attenuation of AGEs production in serum and hepatic tissues were the most outstanding among other doses used in either monotherapy and combined therapy.

Actually, the association of diabetes and inflammation is an item of active research. The role of inflammation was previously demonstrated in the development of T2DM [67] and hyperglycemia was shown to increase the circulating TNF-α and IL-6 [23]. Previous and recent studies on human and animal models exhibit that inflammation and insulin resistance are directly associated with each other throughout the development of T2DM [68, 69]. MET has been reported to decrease inflammatory markers and participate in the reduction of ROS in diabetic patients [70]. Therefore, one of the main objectives of this study was to explore the effect of two doses of Se-NPs and MET on levels of pro-inflammatory cytokines in the liver of T2DM rats. Recently, it was reported that Se-NPs significantly extinguish LPS stimulated NO production and repress the expression of pro-inflammatory cytokines iNOS and TNF-α in a dose-dependent manner [71]. Additionally, inflammation controlled by NF-κB activation, resulting in the overexpression of suppressor proteins socs-3, IL-1β, IL-6, and TNF-α that [72]. In addition, it represses the insulin signaling pathway by down-regulation of IRS1 and AKT expressions, ultimately lead to IR [73, 74]. Activation of NF-κB leads to its translocation from cytoplasm to nucleus, whereas p65, phosphorylated subunit plays the main role in the stimulation of gene expression of certain genes. It is obvious from our results that HFD/STZ increased the hepatic protein levels of p65-NF-κB, and COX-2 combined with an elevation of serum and hepatic IL-1β, TNF-α, IL-6, and iNOS cytokine levels. This is a confirmation of a significant role in the inflammatory process in IR and T2DM. Moreover, the gene expression profile of hepatic inflammatory cytokines and chemokines (il-1α, il-23, mcp-1, socs-3, Ox-LDL-1) were markedly elevated in T2DM rats. On the other hand, monotherapy treatment strategy could markedly exhibit a strong anti-inflammatory effect by reversing those increases in all inflammatory mediators started by strong suppression of p65-NF-κB protein level. Interestingly, the combination therapy of Se-NPs and MET, particular the Se-NPs (0.4 mg/kg) with MET represented a more anti-inflammatory power and showed marked repression in all inflammatory cytokines and chemokines at gene and protein expression levels. Thus, according to the current results, the combined-therapy strategy displayed a synergistic anti-inflammatory power in T2DM.

Although natural products have some potential against chronic diseases as a therapeutic agent, the incorporation of nanoparticles in the therapeutic strategies would provide new avenues to increase their potency against chronic disorders like T2DM. In a comparison of our findings in the current study to our previous studies [52, 68] on some oxidative stress and inflammatory markers in T2DM-induced rats, an improvement was noticed. The results of the current study revealed that the treatment with Se-NPs at dose 0.4 mg/kg alone or combined with MET decreases the MDA levels by 7.0 and 9.0 folds, respectively as well as decreased NO levels by 3.0 and 10.0 folds, respectively in the liver tissues. On the other hand, the treatment with NSO alone or combined with MET decreases MDA levels by 1.9 and 6.0 folds, respectively in the liver and 3-fold in the brain as well as decreased NO levels by 5.3-fold in the liver and 2.0-fold in the brain. Furthermore, Se-NPs at dose 0.4 mg/kg alone or combined with MET decreases the TNF-$\alpha$ levels by 4.0 and 8.0 folds, respectively in the liver, while NSO alone or combined with MET decreased TNF-$\alpha$ levels by 1.5 and 1.2 folds, respectively in the liver and 6.0-fold in the brain.

Furthermore, the anti-diabetic efficiency of the Se-NPs was explored by examining the molecular mechanisms underlying insulin resistance developed in STZ-induced T2DM animals. We assumed that the impairment of the IRS/AKT/GSK-3$\beta$ signaling pathway might have a role in this process. In addition, HFD/STZ model in the present study exhibited lowered levels of phosphorylated IRS1 associated with decreased levels of pAKT1 and pGSK-3$\beta$. The results of our study revealed that HFD/STZ administration induced hyperglycemia and hyperinsulinemia indicating the promotion of T2DM. There was significant repression in IRS/AKT/GSK-3$\beta$ signaling pathway in liver samples of HFD/STZ-induced rats suggesting that HFD/STZ can enhance oxidative stress, diminish the survival of liver cells and induce the insulin resistance. As the insulin signaling pathway controls the transport of glucose in hepatic cells, the dysregulation of IRS/AKT/GSK-3$\beta$ insulin signaling pathway is a key determinant of the glycemic response showed in uncontrolled diabetes. Previous studies demonstrated that the up-regulation of pIRS1 and pAKT might improve the uptake of glucose and reduce blood glucose levels [75]. Therefore, we studied the effect of oral administration of monotherapy and/or combined therapy on insulin signaling pathways in liver cells. In our study, the protein levels of pIRS1/pAKT/pGSK-3$\beta$ increased significantly after mono and combined therapies. However, the effect of combined therapy, especially Se-NPs (0.4 mg/kg) with MET on recovering these protein levels were the most ascendant among the monotherapy. We can reasonably estimate that the role of Se-NPs to induce of increment of selenoproteins may enhance insulin sensitivity by boosting the insulin signaling pathway.

Interestingly, at molecular levels, it has been evidenced that the activation of AMPK ameliorates insulin sensitivity and its activation is suppressed by inflammation and/or IR [76]. AMPK has been stated to regulate IRS1 [77] and AKT [78]. Thus, the activators of AMPK such as MET promote the action of insulin on AKT activation [79] and inhibit cellular inflammation [80]. Our present study has illustrated that the monotherapy and combined therapy increased the pAMPK protein levels. Over and above the enhancement effect of combined-therapy of Se-NPs 0.4 mg/kg with MET showed a significant increment in the pAMPK protein levels compared to other treatments confirming its enhancement of insulin sensitivity via direct regulation of IRS1/AKT/GSK-3$\beta$ pathway. Consequently, based on the above results, Fig 9 schematically illustrates the possible synergizing processes of Se-NPs and MET combination as an integrated multi-functional anti-diabetic therapeutic agent.

## Conclusion

The current study justifies the anti-diabetic potential of oral administration of two doses of Se-NPs (0.1 and 0.4 mg/kg) and/or MET as monotherapy as well as combined therapy can delay

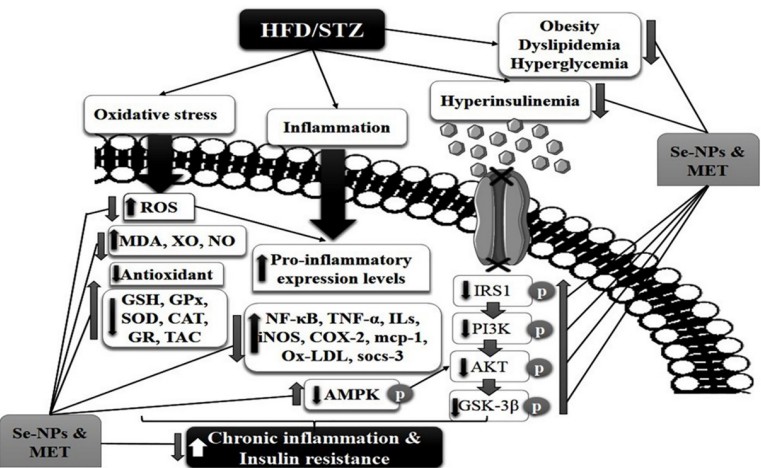

**Fig 9. Mechanisms underlying insulin resistance and inflammatory pathway suppression by selenium nanoparticles and metformin.** They promote the activation of insulin signaling molecules IRS1/AKT/GSK-3β. In addition, they enhance the cellular function via AMPK activation. They alleviate diabetic complications by inhibition of stressful stimuli and improve antioxidant capacity. They improve glucose, insulin homeostasis, and lipid metabolism. They exert their anti-inflammatory efficiencies by interrupting the ROS mediated inflammation and down-regulate all inflammatory mediators. (↑): Up-regulated targets; (↓) Down-regulated targets.

the onset of HFD/STZ-induced T2DM and minimizes the risk of its complications. This occurred possibly via a free radical scavenging effect, improving insulin sensitivity and activating pIRS1/pAKT/pGSK-3β signaling pathway as well as pAMPK. In addition, it ameliorated the damage of hepatic and renal tissues and reduced lipid accumulation caused by elevated glucose levels. Furthermore, it alleviated the expression of pro-inflammatory cytokines and ROS production with restoring the antioxidant capacity. Moreover, Se-NPs combined with MET possessed an outstanding effect on the values of these anti-diabetic biomarkers than either Se-NPs or MET alone. Taken together, Se-NPs with MET may have a natural synergy anti-diabetic activity. Our findings may facilitate the understanding of the novel effects of Se-NPs alone or combined with anti-diabetic drug MET and suggest a newly recognized benefit of Se-NPs in T2DM. Due to high bioavailability and low toxicity, Se-NPs appears to be a unique platform for supplementation, and it provides new opportunities with significant dietary and therapeutic potential for treating patients with T2DM. A further understanding of the molecular mechanisms underlying this function and safety is still in need to evaluate the anti-diabetic activity of Se-NPs.

## Supporting information

**S1 Data. Raw data (the values behind means and standard errors, the values used to build graphs, and the points extracted from images for analysis).**
(XLSX)

**S2 Data.**
(PDF)

## Acknowledgments

The authors like to express their gratitude to Dr. Wesam A. Tawfik for her expert advice on selenium nanoparticles preparation. The authors also would like to thank the staff members of

Department of Biochemistry, Faculty of Science, Alexandria University, especially Dr. Doaa A. Ghareeb for her kind help, technical assistance, and help in some biochemical analyses.

## Author Contributions

**Conceptualization:** Mahmoud Balbaa.

**Data curation:** Shaymaa A. Abdulmalek.

**Formal analysis:** Mahmoud Balbaa.

**Investigation:** Mahmoud Balbaa.

**Methodology:** Shaymaa A. Abdulmalek.

**Supervision:** Mahmoud Balbaa.

**Validation:** Mahmoud Balbaa.

**Writing – original draft:** Shaymaa A. Abdulmalek.

**Writing – review & editing:** Mahmoud Balbaa.

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
