## [Decision Letter · Decision Letter 0]

12 Jun 2019

PONE-D-19-14847

Synergistic effect of nano-selenium and metformin on type 2 diabetes-induced rats: Diabetic complications alleviation through insulin sensitivity, oxidative mediators and inflammatory markers

PLOS ONE

Dear Prof Balbaa,

Thank you for submitting your manuscript to PLOS ONE. After careful consideration, we feel that it has merit but does not fully meet PLOS ONE’s publication criteria as it currently stands. Therefore, we invite you to submit a revised version of the manuscript that addresses the points raised during the review process.

We would appreciate receiving your revised manuscript by Jul 27 2019 11:59PM. To enhance the reproducibility of your results, we recommend that if applicable you deposit your laboratory protocols in protocols.io, where a protocol can be assigned its own identifier (DOI) such that it can be cited independently in the future. For instructions see: http://journals.plos.org/plosone/s/submission-guidelines#loc-laboratory-protocols

We look forward to receiving your revised manuscript.

Kind regards,

Michael Bader

Academic Editor

PLOS ONE

Journal Requirements:

1. In your Material and Methods section we note that you state the following "All animal studies were performed according to the animal protocols approved by the Institutional Ethics Committee of Faculty of Science, Alexandria University, Egypt in accordance with the ethical standards of the Helsinki Declaration of 1975, as revised in 2008."  The Declaration of Helsinki is a set of ethical principles regarding the use of human participants in experimental or medical research. We understand that the submitted work only uses animals and we would therefore kindly ask that you remove the reference fo the Helsinki Declaration.

Reviewers' comments:

Reviewer's Responses to Questions

**Comments to the Author**

1. Is the manuscript technically sound, and do the data support the conclusions?

Reviewer #1: Yes

Reviewer #2: Partly

2. Has the statistical analysis been performed appropriately and rigorously? 

Reviewer #1: Yes

Reviewer #2: Yes

3. Have the authors made all data underlying the findings in their manuscript fully available?

Reviewer #1: Yes

Reviewer #2: Yes

4. Is the manuscript presented in an intelligible fashion and written in standard English?

Reviewer #1: No

Reviewer #2: No

5. Review Comments to the Author

Reviewer #1: This work prepared selenium nanoparticles and investigated their antidiabetic action and synergy with metformin. The study is sufficient and sound in experiments and results. It provides insight into the use of selenium nanoparticles (Se-NPs) to combat type 2 DM. However, some minor revisions are required to be addressed for improving the manuscript. Here, I raised some recommendations to the authors for reference.

1. The title of this article needs to be amended. Type 2 diabetes-induced rats is a wrong expression. HFD/STZ-induced type 2 diabetes rats or type 2 diabetic model rats are more suitable.

2. The functions of selenium as antidiabetic agent should be elaborated in the introduction section. The authors can refer to the latest publication (Guan, B. et al. Selenium as a pleiotropic agent for medical discovery and drug delivery. Int. J. Nanomedicine. 2018, 13: 7473-7490) for citation.

3. In the introduction section, some representative examples using selenium nanoparticles for treatment of diabetes should be contained to highlight the antidiabetic advantages of Se-NPs, for example, Deng, W. et al. Selenium nanoparticles as versatile carriers for oral delivery of insulin: Insight into the synergic antidiabetic effect and mechanism. Nanomedicine. 2017, 13: 1965-1974. Deng, W. et al. Selenium-layered nanoparticles serving for oral delivery of phytomedicines with hypoglycemic activity to synergistically potentiate the antidiabetic effect. Acta Pharmaceutica Sinica B. 2019, 9: 74-86.

4. Regarding the preparation of Se-NPs, why does the coating with dextrin happen after Se-NPs preparation. As a stabilizing agent, polymer is normally concurrently added into the system upon reaction of ionic selenium and reductant.

5. Why is the insulin concentration in HFD/STZ model far higher than the control? Does HFD/STZ induction result in more secretion of insulin? This is abnormal that the insulin level does not reduce but inversely increase in diabetic rats in logicality.

6. The toxicity of Se-NPs is a great concern for its biomedical application. The authors should investigate the cytotoxicity of Se-NPs using cell test or other techniques.

7. The English language of this manuscript should be enhanced. Some expressions are confused in style.

Reviewer #2: - There are many English errors. The work should be revised by an English native speaker.

- The authors can't use "dose-dependent" with two investigated doses only, at list the word used with three doses.

- "HFD was administered" should be "HFD was supplemented".

- "different diabetic complication pathways" vague expression. In this study, the authors focused mainly on the anti-diabetic effect of Se.

- "After 16 weeks of daily oral administration" should be 8 instead of 16.

- Histopathology of liver and pancrease are highly recommended.

- Figure's 2 legend needs more explanation.

- How many assays were performed to obtained the results those used to illustrate fig.3?

- Final BW (µg) ???

6. PLOS authors have the option to publish the peer review history of their article (what does this mean?). If published, this will include your full peer review and any attached files.

Reviewer #1: Yes: Xingwang Zhang

Reviewer #2: No

---

## [Author Response · Author response to Decision Letter 0]

20 Jul 2019

Review Comments to the Author

Reviewer #1: This work prepared selenium nanoparticles and investigated their antidiabetic action and synergy with metformin. The study is sufficient and sound in experiments and results. It provides insight into the use of selenium nanoparticles (Se-NPs) to combat type 2 DM. However, some minor revisions are required to be addressed for improving the manuscript. Here, I raised some recommendations to the authors for reference.

1. The title of this article needs to be amended. Type 2 diabetes-induced rats is a wrong expression. HFD/STZ-induced type 2 diabetes rats or type 2 diabetic model rats are more suitable.

Changed

2. The functions of selenium as antidiabetic agent should be elaborated in the introduction section. The authors can refer to the latest publication (Guan, B. et al. Selenium as a pleiotropic agent for medical discovery and drug delivery. Int. J. Nanomedicine. 2018, 13: 7473-7490) for citation.

This part was inserted to introduction section as one paragraphs in pages 4.

3. In the introduction section, some representative examples using selenium nanoparticles for treatment of diabetes should be contained to highlight the antidiabetic advantages of Se-NPs, for example, Deng, W. et al. Selenium nanoparticles as versatile carriers for oral delivery of insulin: Insight into the synergic antidiabetic effect and mechanism. Nanomedicine. 2017, 13: 1965-1974. Deng, W. et al. Selenium-layered nanoparticles serving for oral delivery of phytomedicines with hypoglycemic activity to synergistically potentiate the antidiabetic effect. Acta Pharmaceutica Sinica B. 2019, 9: 74-86.

This part was inserted to introduction section in the paragraph of pages 4 & 5.

4. Regarding the preparation of Se-NPs, why does the coating with dextrin happen after Se-NPs preparation. As a stabilizing agent, polymer is normally concurrently added into the system upon reaction of ionic selenium and reductant.

In our study, the Se-NPs were coated with 5% dextrin by using a single layer coating method on a magnetic stirrer at ambient temperature instead of adding water. The nanoparticles were diluted by using a dextrin-containing solution and the samples were then washed and dried to increase the particle stability in solutions. In addition, the coating increases the viscosity of the medium, which in turn decreases the interaction between the particles present in the medium and prevent the Se-NPs precipitation. At the refrigeration temperature (4ᵒC), the coated Se-NPs were stable up to 1 year. The preparation and coating of Se-NPs were prepared according to the method described by (Qian Li et al, 2010) with some modifications

References:

1) Qian Li, Tianfeng Chen, Fang Yang, etal. Facile and controllable one-step fabrication of selenium nanoparticles assisted by L-cysteine. Materials Letters, 2010; 64: 614–617.

2) CHAPTER 3 Synthesis, Coating, Characterisation and Stability of Selenium Nanoparticles.

3) SONAM MALHOTRA, NEETU JHA & KRUTIKA DESAI. A superficial synthesis of selenium nanospheres using wet chemical approach. International Journal of Nanotechnology and Application (IJNA), 2014; 3: 7-14 

4) Malhotra S, Welling MN, Mantri SB, Desai K: In vitro and in vivo antioxidant, cytotoxic, and anti-chronic inflammatory arthritic effect of selenium nanoparticles. J Biomed Mater Res B Appl Biomater, 2016; 104: 993–1003.

5. Why is the insulin concentration in HFD/STZ model far higher than the control? Does HFD/STZ induction result in more secretion of insulin? This is abnormal that the insulin level does not reduce but inversely increase in diabetic rats in logicality.

HFD/STZ induction increases the insulin resistance in which more insulin is not recognized with insulin receptor and therefore insulin will be secreted in serum compared to control.

An example of a chemical-induced animal model of diabetes is the HFD/STZ animal model. This model involves a combination of an HFD to bring about hyperinsulinemia, IR, and/or glucose intolerance followed by subsequent injection of a low dose 35 mg/kg STZ, which results in severe reduction in functional-cell mass. The transition from a metabolically healthy state to prediabetes often includes an obese state characterized by hyperinsulinemia, insulin resistance, and dyslipidemia. Further inflammation of the abdominal adipose tissue may worsen the dysfunctional state of the adipocytes, leading to more ectopic fat accumulation, insulin resistance and hyperinsulinemia. Also, this may be due to inability of insulin to act properly on resistant tissues and this resulted in poor glucose utilization. So, β-cells are initially compensated for insulin resistance by elevating insulin secretion. Beside, multiple organs contribute to the development of peripheral insulin resistance via impaired biological response to insulin. Finally, the events leading to β-cell compensatory mechanisms and subsequent β-cell failure in type 2 diabetes involve lipotoxicity and/or glucolipotoxicity, insulin resistance, hyperinsulinemia, and stress, with a modest contribution from low-level inflammation.

References:

1- Søs Skovsø. Modeling type 2 diabetes in rats using high fat diet and streptozotocin. J Diabetes Investig. 2014 Jul; 5(4): 349–358.

2- Saleh S., El-Maraghy N., Reda E. and Barakat W. Modulation of Diabetes and Dyslipidemia in Diabetic Insulin-Resistant Rats by Mangiferin: Role of Adiponectin and TNF-α. An Acad Bras Cienc, 2014. 86(4): 1935-1948.

3- Jia-You Fang, Chih-Hung Lin, Tse-Hung Huang, and Shih-Yi Chuang. In Vivo Rodent Models of Type 2 Diabetes and Their Usefulness for Evaluating Flavonoid Bioactivity. Nutrients. 2019 Mar; 11(3): 530.

4- Anisha A. Gupte, Laurie J. Minze, Maricela Reyes, Yuelan Ren, Xukui Wang, Gerd Brunner, Mohamad Ghosn, Andrea M. Cordero-Reyes, Karen Ding, Domenico Pratico, Joel Morrisett, Zheng-Zheng Shi, Dale J. Hamilton, Christopher J. Lyon, and Willa A. Hsueh. High-Fat Feeding-Induced Hyperinsulinemia Increases Cardiac Glucose Uptake and Mitochondrial Function Despite Peripheral Insulin Resistance. Endocrinology. 2013 Aug; 154(8): 2650–2662.

5- Erin E. Mulvihill, Emma M. Allister, Brian G. Sutherland, Dawn E. Telford, Cynthia G. Sawyez, Jane Y. Edwards, Janet M. Markle, Robert A. Hegele, and Murray W. Huff. Naringenin Prevents Dyslipidemia, Apolipoprotein B Overproduction, and Hyperinsulinemia in LDL Receptor–Null Mice With Diet-Induced Insulin Resistance. Diabetes. 2009 Oct; 58(10): 2198–2210.

6- Balbaa M, El-Zeftawy M, Ghareeb D, Taha N, Mandour AW. Nigella sativa Relieves the Altered Insulin Receptor Signaling in Streptozotocin-Induced Diabetic Rats Fed with a High-Fat Diet. Oxid Med Cell Longev. 2016;2016:2492107.

6. The toxicity of Se-NPs is a great concern for its biomedical application. The authors should investigate the cytotoxicity of Se-NPs using cell test or other techniques.

The determination of the parameters of hepatic function (Table 2) and renal function (Table 3) are important toxicological parameters as explained in page 14 in the subtitle” Serum liver and kidney function parameters”. In addition, an experiment was run to investigate the cytotoxicity of Se-NPs in a hepatocyte cell line. This investigation was inserted in “Materials and Methods” in the subtitle “Cytotoxicity study” (page7 & 8) and “Results” in the subtitle “Cytotoxicity Analysis” (page 14). Also, it was discussed in the first paragraph in “Discussion” (page 19 & 20). Moreover, there is no detected hepatic and renal toxicity by Se-NPs in the tested animals as shown in Tables 3 &4.

7. The English language of this manuscript should be enhanced. Some expressions are confused in style.

The whole file of the manuscript was subjected to intensive language revision for the contextual spelling, grammar and sentence structure. Minor mistakes were found and corrected (grey-labeled).

Reviewer #2: - There are many English errors. The work should be revised by an English native speaker.

The whole file of the manuscript was subjected to intensive language revision for the contextual spelling, grammar and sentence structure. Minor mistakes were found and corrected (grey-labeled).

- The authors can't use "dose-dependent" with two investigated doses only, at list the word used with three doses.

Corrected

All the expressions of “dose-dependent” were deleted from the test, table legends and figure caption except what explained in page 21 in “Discussion”, reference 62.

- "HFD was administered" should be "HFD was supplemented".

Corrected in abstract and pages 9 & 10.

- "different diabetic complication pathways" vague expression. In this study, the authors focused mainly on the anti-diabetic effect of Se.

“insulin sensitivity, oxidative mediators and inflammatory markers” replaces the mentioned expression.

- "After 16 weeks of daily oral administration" should be 8 instead of 16.

Corrected

"After 8 weeks of daily oral administration" 

- Histopathology of liver and pancreas are highly recommended.

Since our target project has focused on the study of different signaling and inflammatory parameters, the histopathological studies of liver and pancreas were not in our plan. The tissues were not kept in formalin for the histopathological studies and we will consider it in the future in an extended study. 

- Figure's 2 legend needs more explanation.

Changed to 

Fig. 2. Representative Transmission Electron Microscopy analysis of prepared Se-NPs showing their size and shape.

- How many assays were performed to obtain the results those used to illustrate fig.3?

Five assays were performed to determine In vitro antioxidant bioactivities of prepared Se-NPs 

1- DPPH radical scavenging activity

2- Nitric oxide radical scavenging activity

3- Hydrogen peroxide scavenging activity

4- Total antioxidant capacity assay

5- Reducing power assay

- Final BW (µg) ???

Corrected as Final BW (g). It is the body weight at the end of the 16 weeks expressed in grams.

---

## [Editor Report · Decision Letter 1]

24 Jul 2019

Synergistic effect of nano-selenium and metformin on type 2 diabetic rat model: Diabetic complications alleviation through insulin sensitivity, oxidative mediators and inflammatory markers

PONE-D-19-14847R1

Dear Dr. Balbaa,

We are pleased to inform you that your manuscript has been judged scientifically suitable for publication and will be formally accepted for publication once it complies with all outstanding technical requirements.

With kind regards,

Michael Bader

Academic Editor

PLOS ONE
---

## [Editor Report · Acceptance letter]

16 Aug 2019

PONE-D-19-14847R1 

Synergistic effect of nano-selenium and metformin on type 2 diabetic rat model: Diabetic complications alleviation through insulin sensitivity, oxidative mediators and inflammatory markers 

Dear Dr. Balbaa:

I am pleased to inform you that your manuscript has been deemed suitable for publication in PLOS ONE. Congratulations! Your manuscript is now with our production department. 

With kind regards,

on behalf of

Prof. Michael Bader 

Academic Editor

PLOS ONE